# VTBench: Evaluating Visual Tokenizers for Autoregressive Image Generation

## Abstract

Autoregressive (AR) models have recently shown strong performance in image generation, where a critical component is the visual tokenizer (VT) that maps continuous pixel inputs to discrete token sequences. The quality of the VT largely defines the upper bound of AR model performance. However, current discrete VTs fall significantly behind continuous variational autoencoders (VAEs), leading to degraded image reconstructions and poor preservation of details and text. Existing benchmarks focus on end-to-end generation quality, without isolating VT performance. To address this gap, we introduce VTBench, a comprehensive benchmark that systematically evaluates VTs across three core tasks: Image Reconstruction, Detail Preservation, and Text Preservation, and covers a diverse range of evaluation scenarios. We systematically assess state-of-the-art VTs using a set of metrics to evaluate the quality of reconstructed images. Our findings reveal that continuous VAEs produce superior visual representations compared to discrete VTs, particularly in retaining spatial structure and semantic detail. In contrast, the degraded representations produced by discrete VTs often lead to distorted reconstructions, loss of fine-grained textures, and failures in preserving text and object integrity. Furthermore, we conduct experiments on GPT-4o image generation and discuss its potential AR nature, offering new insights into the role of visual tokenization. We release our benchmark and codebase publicly to support further research and call on the community to develop strong, general-purpose open-source VTs.

## 1 Introduction

Large Language Models (LLMs) have demonstrated impressive generalization across a wide range of task, including reasoning (Ning et al., 2024; Plaat et al., 2024), question answering (Zhang et al., 2024; Sima et al., 2024) and text generation (Touvron et al., 2023; DeepSeek-AI et al., 2024; Jiang et al., 2024). Recent advances suggest that integrating visual understanding and generation into the LLM framework could lead to unified, general-purpose multimodal models (Fan et al., 2024; Lu et al., 2024; Team, 2024; Xie et al., 2024; Ge et al., 2024).

**Visual Tokenizer.** In diffusion-based image generation, images are typically compressed into a continuous latent space using a variational autoencoder (VAE), allowing the model to operate in a lower-dimensional but still continuous domain (Ho et al., 2020; Rombach et al., 2021). However, when integrating visual understanding and generation into LLMs, images must be converted into discrete token sequences, similar to word or subword tokens in natural language processing (Esser et al., 2021; Chang et al., 2022; Wang et al., 2023). Figure 2 illustrates various visual tokenizer architectures and the autoregressive modeling used for image generation.

**Vector Quantized Tokenization.** To enable such tokenization of visual inputs, Vector Quantized Variational Autoencoder (VQ-VAE) is proposed to encode images into a continuous latent space and then map each latent vector to the nearest entry in a learned codebook, producing a discrete index per spatial location (van den Oord et al., 2017; Esser et al., 2021). However, scaling the codebook often leads to codebook collapse, where only a small portion of entries are used, reducing representational capacity. Additionally, nearest-neighbor search introduces computational inefficiency during generation (Yu et al., 2024a; Han et al., 2024). To address these limitations, Lookup-Free Quantization (LFQ) eliminates the embedding lookup by projecting continuous features into a binary latent

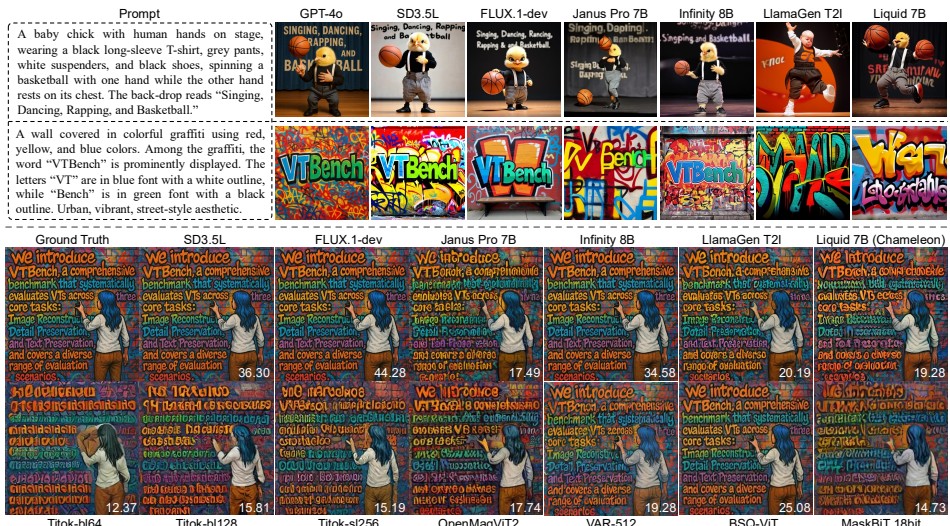

Figure 1: Image generation and reconstruction across different models. **(Top)** Images generated from prompts using various models. **(Bottom)** Reconstructions of the ground truth image using VTs from different models. PSNR ↑ values (shown in white) indicates reconstruction fidelity.

space, enabling discrete tokenization without explicit search (Yu et al., 2024a). To constrain error and enhance stability, Binary Spherical Quantization (BSQ) introduces $\ell_2$ normalization, smoothing the representation and facilitating optimization (Zhao et al., 2024).

**Next-Scale Tokenization.** However, AR modeling with existing visual tokenizers often violates the unidirectional dependency assumption and disrupts spatial locality due to the flattening of 2D token grids. To overcome these limitations, Tian et al. (2024) proposed Visual Autoregressive Modeling (VAR), which encodes images into multiple levels of tokens by progressively quantizing the residual information at each scale. This hierarchical tokenization preserves spatial structure and allows the model to capture fine-grained details in a coarse-to-fine manner. In Figure 2, we refer to this architecture as "Residual Next-Scale VAE". Building on this foundation, Infinity enhances visual tokenization by integrating BSQ, enabling extremely large vocabularies and efficient scaling. This allows Infinity to generate high-resolution images with unprecedented detail (Han et al., 2024).

Despite the growing number of AR models for image generation (Li et al., 2024; Bie et al., 2025; Wu et al., 2024b), most existing methods are still limited to relatively simple datasets such as ImageNet and fall significantly behind diffusion models in terms of generation quality and detail preservation. As shown in Figure 1(Top), which presents outputs from several SOTA text-to-image models prompted with highly detailed descriptions, only the images generated by diffusion models (SD3.5L (Rombach et al., 2022) and FLUX.1-dev (Labs, 2024)) and GPT-4o (architecture currently undisclosed (Yan et al., 2025; Li et al., 2025)) exhibit high-quality synthesis. We hypothesize that the visual tokenizer plays a critical role in this quality gap. To investigate this, we conduct a reconstruction experiment using the images generated by GPT-4o as the ground truth. As shown in Figure 1(Bottom), we reconstruct the ground truth image using different VTs from various AR models. The results reveal substantial information loss, including blurred or unreadable text, loss of fine-grained visual details, and noticeable structural distortions. These failures indicate that current VTs struggle to generate accurate and expressive latent representations, which limits the overall image generation quality of AR models.

*Why Benchmarking VT Matters?* The reconstruction failures observed in Figure 1 highlight a fundamental issue: current visual tokenizers often fail to preserve fine-grained details and semantic integrity during the quantization process (Yu et al., 2024a; Zhao et al., 2024). This failure propagates through the entire AR generation pipeline, ultimately degrading image quality regardless of the downstream model's capacity. Despite their central role, existing evaluation protocols focus almost exclusively on end-to-end generation quality, without isolating the contribution or limitations of the VT itself. This leaves several critical issues unaddressed: (1) **Lack of VT-Specific Evaluation:** The performance of the VT often determines the upper bound of AR model quality (Yu et al., 2024a). Yet, most VTs are only evaluated on limited datasets like ImageNet (Russakovsky et al., 2015), and there is a lack of dedicated benchmarks designed specifically to measure VT effectiveness across diverse

Figure 2: Overview of visual tokenizer architectures and integration with AR image generation.

scenarios. (2) **Benchmark Misalignment:** Existing benchmarks evaluate overall image generation (e.g., ImageNet (Russakovsky et al., 2015), GenEval (Ghosh et al., 2023), T2i-compbench (Huang et al., 2023; 2025)), rather than isolating the visual tokenizer contribution, making it difficult to diagnose or improve this key component. (3) **Inadequate Evaluation Metrics:** Commonly used metrics such as FID are insufficient to capture fine-grained failures like high-frequency detail loss or incorrect text reconstruction, which are essential for many multimodal and LLM-centric tasks.

To address these issues, in this paper, we introduce VTBench, a comprehensive benchmark specifically designed to evaluate visual tokenizers across a broad range of tasks, datasets, and conditions. VTBench provides a systematic framework for understanding and improving VTs as standalone components in AR image generation pipelines. Through extensive experiments on a wide range of VTs used in SOTA AR models, we uncover the following key findings: (1) Existing discrete VTs fall significantly behind continuous VAEs used in diffusion models, particularly in terms of reconstruction quality, detail preservation, and text accuracy. (2) None of the existing VTs can robustly handle arbitrary resolutions, unlike continuous VAEs, although VAR (Tian et al., 2024) and Infinity (Han et al., 2024) are restricted to a fixed set of predefined input sizes. (3) Many AR models (e.g., Chameleon (Team, 2024), Liquid (Wu et al., 2024a), Anole (Chern et al., 2024)) reuse the same open-source VT, yet there is currently no strong, general-purpose VT available for reuse, highlighting the lack of a reliable, high-quality open-source solution in this space.

**Contributions.** The main contributions of this paper are:

- We introduce VTBench, a high-quality and comprehensive benchmark designed specifically for evaluating VTs in the context of AR image generation.
- We design three tasks: Image Reconstruction, Detail Preservation, and Text Preservation, collectively providing a multi-faceted framework for assessing visual tokenizers. These tasks cover diverse evaluation aspects, including high-resolution inputs, multilingual text scenarios (Chinese, Korean, Japanese, Hindi), and varying-resolution conditions.
- We conduct extensive experiments on VTs used in SOTA AR models, including continuous VAE, GPT4o, VQVAE, etc. Our evaluation is designed to assess both image quality and text preservation through a diverse set of quantitative metrics.
- We provide an in-depth comparison and discussion of current VTs and contrast their behavior with the emerging capabilities of GPT-4o's VT. We identify fundamental gaps and discuss directions for future tokenizer development.
- We open-source both the codebase and the VTBench dataset to foster further research in visual tokenization for autoregressive image generation. Our codebase is designed to be lightweight and easy to run, requiring minimal setup and no complex configuration.

## 2 BACKGROUND

In this section, we introduce image generation methods, focusing on both diffusion and AR models. We then discuss the role of visual tokenizers in AR pipelines, reviewing several architectures including VQ-VAE, LFQ-VAE, BSQ-VAE, and Residual Next-Scale VAE, as illustrated in Figure 2. Finally, we highlight recent advances in image generation with GPT-4o, which motivate the need for deeper evaluation of visual tokenization quality.

### 2.1 IMAGE GENERATION: DIFFUSION VS. AUTOREGRESSIVE MODELS

Modern image generation methods are mainly dominated by two families: diffusion models (Croitoru et al., 2023; Yang et al., 2024) and AR models (Fan et al., 2025; Chen et al., 2025a; Yu et al., 2024b). Diffusion models learn to iteratively denoise a sample from pure noise to a realistic

image using a learned reverse diffusion process, such as Stable Diffusion (Rombach et al., 2022), which have demonstrated SOTA performance in high-quality image synthesis. In contrast, AR models decompose image generation into a sequence modeling problem, where an image is represented as a sequence of discrete tokens predicted one at a time (Ma et al., 2025; Luo et al., 2024; Yu et al., 2023), as shown in Figure 2(f). AR models benefit from the scaling properties of LLMs and enable seamless integration with multimodal pipelines. However, AR image generation relies heavily on high-quality visual tokenization to convert pixel data into token sequences and reconstruct images from these tokens, making the design of effective visual tokenizers a central challenge in AR-based image generation.

## 2.2 VISUAL TOKENIZERS

To enable AR models to process image inputs, images must be converted into discrete token sequences. This is achieved via visual tokenizers that compress high-dimensional image data into compact, symbolic representations. In this section, we briefly introduce several representative tokenization approaches. We briefly introduce representative tokenization approaches here; a detailed version with mathematical formulations is included in Appendix B.

- **Continuous VAE.** In diffusion models, a common approach is to use a continuous Variational Autoencoder (VAE) (Rombach et al., 2022; Peebles & Xie, 2023) as a feature compressor as shown in Figure 2(a). These VAEs encode the input image into a continuous latent space, typically using convolutional encoders and decoders, enabling high-quality image reconstructions.
- **VQ-VAE.** Vector Quantized Variational Autoencoder (VQ-VAE) (van den Oord et al., 2017) is one of the earliest and most widely adopted discrete visual tokenizers. It encodes an image into a latent feature map and then quantizes each spatial location to the closest vector in a learned codebook. The resulting discrete indices form a token grid that can be used in AR models.
- **LFQ-VAE.** Lookup-Free Quantization (LFQ) (Yu et al., 2024a) eliminates the nearest-neighbor lookup and codebook by projecting features into a binary latent space by a learnable mapping.
- **BSQ-VAE.** Binary Spherical Quantization (BSQ) (Zhao et al., 2024) extends LFQ by applying $\ell_2$ normalization to the latent features before quantization. This constrains the representation to a unit hypersphere, effectively reducing quantization error and leading to smoother latent spaces. BSQ allows for finer-grained tokenization while maintaining efficient binary encoding, which is beneficial for downstream tasks sensitive to visual detail.
- **Residual Next-Scale VAE.** applies hierarchical tokenization by encoding images in a coarse-to-fine manner. It quantizes residual information across multiple spatial scales, enabling preservation of structure and fine details, especially in high-resolution settings (Tian et al., 2024).

## 2.3 GPT-4O IMAGE GENERATION

OpenAI recently introduced new image generation and editing capabilities in GPT-4o (OpenAI, 2025), showcasing remarkable performance in both tasks. Although the precise architecture and workflow of GPT-4o remain undisclosed, recent studies suggest that the model may employ an autoregressive backbone in conjunction with a diffusion-based decoder for image synthesis(Yan et al., 2025; Li et al., 2025). In this paper, we present extensive experiments on GPT-4o to investigate its architecture and workflow, with a focus on how the visual tokenizer influences image quality.

## 3 VTBENCH: TASK SETTINGS & EVALUATION METRICS

To systematically evaluate VTs in AR image generation pipelines, we propose VTBench, a comprehensive benchmark that isolates and diagnoses the capabilities and limitations of VTs across three critical tasks: (1) Image Reconstruction, (2) Detail Preservation, and (3) Text Preservation. Each task is designed to stress different aspects of tokenization quality, using diverse data conditions.

## 3.1 TASK 1: IMAGE RECONSTRUCTION

This task evaluates the fundamental ability of a VT to reconstruct an image from its tokenized representation. As the interface between high-dimensional pixel inputs and the discrete token sequences consumed by AR models, the VT plays a critical role in determining the upper bound of image generation quality. If the tokenizer discards or distorts essential visual information, such as fine-grained

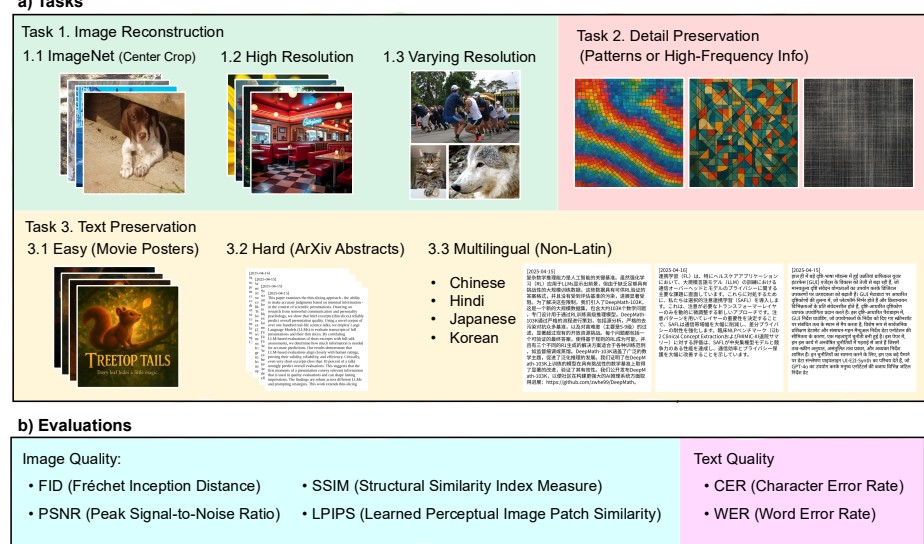

Figure 3: Overview of VTBench construction. (a) VTBench consists of three core tasks for evaluating visual tokenizers. (b) Evaluation include both image quality metrics and text-specific metrics.

structure, object boundaries, or spatial layout – this loss is irreversible, regardless of the strength of the downstream generative model.

To isolate the intrinsic capacity of each VT, we assess image reconstruction performance independent of generation. Specifically, we evaluate three settings designed to test robustness across resolution and scale: (1) ImageNet (model-specific input size), (2) High Resolution (1024 × 1024 inputs), and (3) Varying Resolution (images with diverse, unconstrained dimensions), as illustrated in Figure 3(a).

**ImageNet.** We begin with the validation subset of the standard ImageNet-1k dataset (Russakovsky et al., 2015), containing 50,000 images. This benchmark serves as a canonical reference point, as most open-source VTs are evaluated on it. The task assesses whether the VT preserves general semantics and structural integrity in a typical low-resolution setting. In this subtask, all images are center-cropped and resized to model-specific input sizes as listed in Table 1.

**High Resolution.** To evaluate scalability, we synthesize 100 high-resolution images (1024 × 1024) using GPT-4o, and assess the quality of their reconstructions. This setting exposes limitations in the VT's ability to preserve visual fidelity across larger spatial extents, where coarse quantization or resolution mismatches often degrade reconstruction quality. The full high-resolution image synthesis process is detailed in Appendix C.

**Varying Resolution.** In real-world scenarios, image resolutions vary widely, posing challenges for models that assume fixed-size inputs. To evaluate the robustness of VTs under such conditions, we use a mixed-resolution dataset, specifically the test subset of DIV2K (Agustsson & Timofte, 2017), which includes 100 high-quality images with a broad range of dimensions. Unlike continuous VAEs, most discrete tokenizers are not resolution-agnostic and require fixed-size inputs. This task quantifies how such constraints affect reconstruction accuracy and the model's flexibility in handling diverse input sizes, with a detailed architectural analysis presented in Appendix E.

### 3.2 TASK 2: DETAIL PRESERVATION

While overall reconstruction quality is important, many downstream tasks, such as object recognition, editing, and captioning, depend on the preservation of fine-grained visual details. These include textures, facial features, edges, and small objects that may occupy only a few pixels. Such details are often the first to be degraded or lost during quantization, especially when the codebook or token representation lacks sufficient expressiveness. A VT that fails to preserve these features will fundamentally limit the quality and realism of generated images. Therefore, this task focuses on measuring how well VTs retain high-frequency information crucial for perceptual fidelity. We follow the same high-resolution synthesis procedure described in Appendix C, using GPT-4o to generate 100 images rich in detailed, high-frequency semantic content. We provide a more detailed definition of

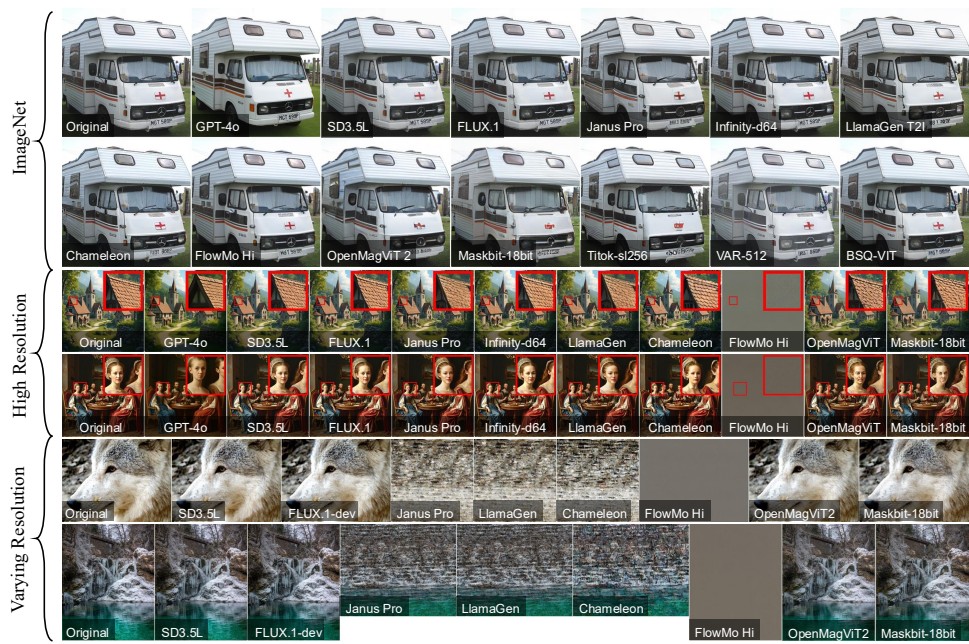

Figure 4: Examples of task 1: (1) ImageNet, (2) High Resolution and (3) Varying Resolution. FlowMo Hi produces corrupted images in High Resolution and Varying Resolution, while Janus Pro, LlamaGen, and Chameleon generate images with incorrect resolution and distorted semantics.

"visual details" and quantify that images in Task 2 contain more fine-grained information than other tasks in Appendix D.

### 3.3 TASK 3: TEXT PRESERVATION

Text is a uniquely challenging and critical element in many real-world images, especially in domains such as documents, signage, user interfaces, and multimodal reasoning. Unlike general textures, text requires pixel-level precision, minor distortions can render characters unreadable, break words, or alter semantics entirely. Furthermore, text carries symbolic meaning that must be preserved for models to support language-image alignment, OCR, or instruction-following. However, most VTs are not optimized for symbolic preservation, especially in multilingual or high-density settings. This task evaluates how VTs preserve textual content under varying complexity and linguistic diversity.

**Movie Posters (Easy).** In this subtask, we synthesize 100 images of movie posters. Using GPT-4o, we first generate descriptions for 100 fictional movies along with corresponding slogans, and then produce a poster for each movie. Each poster features a prominent title in large text, a slogan in smaller text, and a background that visually reflects the movie's theme or genre. Details of synthesis are provided in Appendix C. The dataset spans a wide variety of movie types, providing diverse yet structured visual compositions. A strong VT should preserve both the overall image quality and the legibility of embedded text. Example posters are shown in Figure 3.

**ArXiv Abstracts (Hard).** In addition to the movie posters, we introduce a more challenging subtask consisting of 100 images that render academic paper abstracts. These abstracts feature long sentences, dense layouts, varied font styles, and small font sizes, making this setting particularly demanding. The goal of this task is to evaluate the tokenizer's ability to preserve fine-grained textual content and maintain layout fidelity under complex visual conditions. To synthesize this dataset, we retrieved abstracts from papers published on April 16, 2025 to ensure that the rendered text images were not included in the training data of any existing VT models.

**Multilingual (Non-Latin).** To evaluate the cross-lingual robustness of visual tokenizers, we construct a multilingual benchmark consisting of non-Latin scripts, including Chinese, Hindi, Japanese, and Korean. Specifically, we translate the ArXiv abstract texts into each target language using GPT-4o, and then render the translated content into images following the same formatting and layout procedures as in the English version. For each language, we generate 100 text-rich images that reflect real-world typographic complexity. This subtask assesses whether VTs can preserve diverse character sets, linguistic structures, and writing systems that differ from Latin-based scripts.

Table 1: Evaluation of VTs across three reconstruction settings in Task 1.

| Method | Params | Type | ImageNet | | | | | High Resolution | | | | | Varying Resolution | | | |
|---|---|---|---|---|---|---|---|---|---|---|---|---|---|---|---|---|
| | | | Resolution | PSNR↑ | SSIM↑ | LPIPS↓ | FID↓ | Resolution | PSNR↑ | SSIM↑ | LPIPS↓ | FID↓ | PSNR↑ | SSIM↑ | LPIPS↓ | FID↓ |
| FlowMo Lo | 945M | LFQ | 256 × 256 | 20.2232 | 0.5878 | 0.1031 | 0.8227 | 1024 × 1024 | 11.3631 | 0.3644 | 0.6913 | 465.3528 | 11.4990 | 0.2854 | 0.7379 | 442.0812 |
| FlowMo Hi | 946M | LFQ | 256 × 256 | 22.4923 | 0.7103 | 0.0684 | 0.4834 | 1024 × 1024 | 11.3077 | 0.3874 | 0.6937 | 439.7494 | 11.4571 | 0.2862 | 0.7353 | 412.3167 |
| MaskBiT 16bit | 54M | LFQ | 256 × 256 | 19.6448 | 0.5259 | 0.1246 | 1.1473 | 1024 × 1024 | 28.6892 | 0.9074 | 0.0313 | 24.2461 | 23.9538 | 0.8037 | 0.0405 | 22.4535 |
| MaskBiT 18bit | 55M | LFQ | 256 × 256 | 19.7695 | 0.5352 | 0.1197 | 1.0334 | 1024 × 1024 | 28.9353 | 0.9135 | 0.0295 | 23.3024 | 23.9289 | 0.8070 | 0.0394 | 23.3959 |
| Titok-l32 | 641M | VQ | 256 × 256 | 15.0997 | 0.3533 | 0.2922 | 1.8181 | 1024 × 1024 | - | - | - | - | - | - | - | - |
| Titok-b64 | 204M | VQ | 256 × 256 | 16.1520 | 0.3903 | 0.2285 | 1.3542 | 1024 × 1024 | - | - | - | - | - | - | - | - |
| Titok-s128 | 83M | VQ | 256 × 256 | 16.7386 | 0.4123 | 0.1898 | 1.4186 | 1024 × 1024 | - | - | - | - | - | - | - | - |
| Titok-bl64 | 390M | VQ | 256 × 256 | 17.3930 | 0.4321 | 0.1926 | 1.7696 | 1024 × 1024 | - | - | - | - | - | - | - | - |
| Titok-bl128 | 390M | VQ | 256 × 256 | 18.4675 | 0.4877 | 0.1468 | 1.2328 | 1024 × 1024 | - | - | - | - | - | - | - | - |
| Titok-sl256 | 330M | VQ | 256 × 256 | 19.6375 | 0.5514 | 0.1108 | 0.8112 | 1024 × 1024 | - | - | - | - | - | - | - | - |
| OpenMagViT2 | 115M | LFQ | 256 × 256 | 20.0090 | 0.5786 | 0.1028 | 1.0598 | 1024 × 1024 | 30.7770 | 0.9344 | 0.0232 | 18.6649 | 24.2372 | 0.8133 | 0.0383 | 20.3403 |
| LlamaGen ds8 | 70M | VQ | 256 × 256 | 22.4021 | 0.6995 | 0.0592 | 0.4597 | 1024 × 1024 | 33.5435 | 0.9693 | 0.0112 | 10.1195 | 13.6772 | 0.2735 | 0.5401 | 330.8865 |
| BSQ-VIT | 174M | RBS | 256 × 256 | **25.8646** | **0.8331** | **0.0359** | 0.4586 | 1024 × 1024 | - | - | - | - | - | - | - | - |
| VAR-256 | 109M | RVQ | 256 × 256 | 20.3693 | 0.6035 | 0.0933 | 0.9080 | 1024 × 1024 | - | - | - | - | - | - | - | - |
| Janus Pro 1B/7B | 72M | VQ | 384 × 384 | 22.4485 | 0.6793 | 0.0819 | 0.7111 | 1024 × 1024 | 29.0476 | 0.9208 | 0.0299 | 24.4655 | 13.1547 | 0.2342 | 0.5137 | 324.0125 |
| Chameleon | 68M | VQ | 512 × 512 | 23.5837 | 0.7164 | 0.0672 | 0.8061 | 1024 × 1024 | 29.3743 | 0.9062 | 0.0279 | 22.6725 | 12.7700 | 0.2051 | 0.5206 | 331.3999 |
| LlamaGen ds16 | 72M | VQ | 512 × 512 | **24.2199** | 0.7568 | **0.0630** | **0.5441** | 1024 × 1024 | 28.6424 | 0.9148 | 0.0328 | 25.0529 | 13.3258 | 0.2413 | 0.5171 | 327.8920 |
| LlamaGen ds16 T2I | 72M | VQ | 512 × 512 | 24.0192 | 0.7493 | 0.0647 | 0.5528 | 1024 × 1024 | 29.0061 | 0.9206 | 0.0300 | 24.5831 | 13.1604 | 0.2344 | 0.5137 | 324.4282 |
| VAR-512 | 109M | RVQ | 512 × 512 | 22.3828 | **0.7636** | 0.0727 | 0.7719 | 1024 × 1024 | - | - | - | - | - | - | - | - |
| Infinity-d32 | 110M | RBSQ | 1024 × 1024 | 33.4850 | 0.9582 | 0.0109 | 0.1002 | 1024 × 1024 | 35.5645 | 0.9732 | 0.0080 | 8.3090 | - | - | - | - |
| Infinity-d64 | 110M | RBSQ | 1024 × 1024 | 36.0010 | 0.9766 | 0.0067 | 0.0511 | 1024 × 1024 | 37.5937 | 0.9823 | 0.0053 | 5.8734 | - | - | - | - |
| SD3.5L | 83M | Continuous | 1024 × 1024 | 38.8208 | 0.9836 | 0.0013 | 0.0121 | 1024 × 1024 | 38.4653 | 0.9839 | 0.0012 | 0.9787 | 30.6413 | 0.9507 | 0.0075 | 3.1084 |
| FLUX.1-dev | 83M | Continuous | 1024 × 1024 | **41.6134** | **0.9932** | **0.0006** | **0.0050** | 1024 × 1024 | **41.4870** | **0.9921** | **0.0005** | **0.4466** | **30.9502** | **0.9594** | **0.0068** | **2.7123** |
| GPT-4o | - | - | 1024 × 1024 | 19.3828 | 0.5769 | 0.1463 | 39.5056 | 1024 × 1024 | 15.3835 | 0.4642 | 0.2077 | 78.4113 | - | - | - | - |

## 3.4 Evaluation Metrics

To comprehensively evaluate VTs across the three core tasks of VTBench, we adopt a combination of standard and task-specific metrics to capture different dimensions of quality.

**Image Quality.** To assess how well a VT reconstructs the original image from its tokenized representation, we employ four metrics: (1) **PSNR (Peak Signal-to-Noise Ratio):** Measures pixel-level fidelity between original and reconstructed images. Higher values indicate better reconstruction. (2) **SSIM (Structural Similarity Index):** Quantifies structural similarity by comparing luminance, contrast, and texture, and is more perceptually aligned than PSNR. (3) **LPIPS (Learned Perceptual Image Patch Similarity):** A perceptual similarity metric based on features from pretrained networks (Zhang et al., 2018), correlating well with human judgment. (4) **FID (Fréchet Inception Distance):** Measures distributional distance between reconstructed images in the feature space of an Inception network. Lower FID indicates higher quality.

**Text Quality.** Preserving text in images is critical for multimodal reasoning and OCR-related tasks. To quantify this, we apply OCR to the reconstructed images and compare the results to the OCR result of the original image using: (1) **CER (Character Error Rate):** The Levenshtein distance between predicted and ground truth characters, normalized by total character count. (2) **WER (Word Error Rate):** Similar to CER but computed over word sequences. It is particularly sensitive to segmentation and spelling accuracy. For OCR-based evaluation, we use Gemma 3 (Kamath et al., 2025), a SOTA multimodal model, to extract content from original and reconstructed images to calculate WER and CER. Details of OCR and WER/CER computation are included in Appendix F.

These metrics allow us to analyze not only how much information is retained by the VT, but also what types of information (structural, perceptual, or symbolic) are lost in the tokenization process.

## 4 Experimental Results & Analysis

**Models.** We evaluate a diverse set of VTs spanning multiple quantization paradigms, including VQ, LFQ, BSQ, RVQ, and RBSQ. Our benchmark covers a wide range of SOTA models such as FlowMo (Sargent et al., 2025), MaskBiT (Weber et al., 2024), Titok (Yu et al., 2024b), Open-MagViT2 (Luo et al., 2024), LlamaGen (Sun et al., 2024), BSQ-ViT (Zhao et al., 2024), VAR (Tian et al., 2024), Janus Pro (Chen et al., 2025b), Chameleon (Team, 2024), and Infinity (Han et al., 2024). In our experiments, we specifically evaluate the VT component in isolation, without modifying or including the downstream generation models. This design choice allows us to focus purely on the tokenizer's ability to preserve visual information. For comparison, we also include continuous VAEs used in diffusion models (e.g., SD3.5L (Rombach et al., 2022), FLUX.1-dev (Labs, 2024)) and provide results from GPT-4o (OpenAI, 2025) as a reference, although its tokenizer architecture remains undisclosed (Yan et al., 2025). For GPT-4o, we use the prompt "*Please recreate the exact same image without any alterations and preserve the original resolution.*" to reconstruct images. Table 1 summarizes all evaluated models with their parameter counts and quantization types. Detailed experimental settings and environments are reported in Appendix H.

## 4.1 Task 1: Image Reconstruction

In this task, we evaluate image reconstruction across three subtasks: (1) ImageNet: Images are center-cropped and resized to match each model's required input size. (2) High Resolution: All images are set to a fixed size of 1024 × 1024, and we retain this resolution for all models without

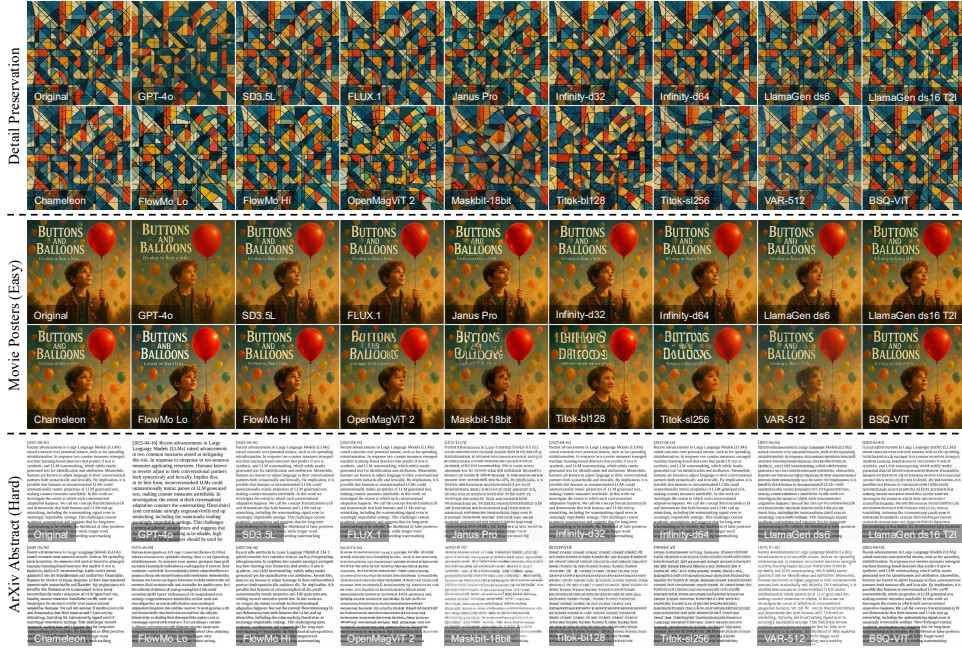

Figure 5: Visualize qualitative results of detail preservation and text preservation.

resizing. (3) Varying Resolution: This subset contains images with diverse, non-uniform resolutions, and we preserve their original sizes for evaluation. We assess reconstruction quality using PSNR, SSIM, LPIPS, and FID. The results, presented in Table 1, reveal a substantial performance gap between all discrete visual tokenizers and continuous VAEs, with the latter consistently achieving better fidelity and perceptual quality. Notably, results from the High Resolution and Varying Resolution settings highlight that most VTs are limited to model-specific input sizes and fail to generalize to arbitrary resolutions, unlike continuous VAEs, which naturally support flexible image dimensions.

Figure 4 presents qualitative examples illustrating reconstruction quality across different VTs. We observe consistent semantic degradations in discrete VT outputs. For example, red crosses on ambulances are missing or blurred (ImageNet), rooftops are distorted, and facial details, and expressions are noticeably altered (High Resolution). In the Varying Resolution, background textures and object boundaries become heavily corrupted. While models such as Janus Pro, LlamaGen, Chameleon, and FlowMo do not raise runtime errors, they fail to generate correct semantic content, highlighting the inherent limitations of current VTs, especially under high or non-regular resolutions.

Table 2: Evaluation of VTs on task 2.

| Method | Image Size | PSNR↑ | SSIM↑ | LPIPS↓ | FID↓ |
|---|---|---|---|---|---|
| FlowMo Lo | 256 × 256 | 17.0642 | 0.3578 | 0.1344 | 64.2895 |
| FlowMo Hi | 256 × 256 | 19.0346 | 0.5578 | 0.0845 | 44.5536 |
| MaskBiT 16bit | 256 × 256 | 17.2804 | 0.2644 | 0.1803 | 98.5133 |
| MaskBiT 18bit | 256 × 256 | 17.3603 | 0.2728 | 0.1745 | 91.3108 |
| Titok-l32 | 256 × 256 | 15.0325 | 0.1609 | 0.3474 | 169.2162 |
| Titok-b64 | 256 × 256 | 15.3827 | 0.1666 | 0.2767 | 124.9069 |
| Titok-s128 | 256 × 256 | 15.6532 | 0.1828 | 0.2316 | 107.7550 |
| Titok-bl64 | 256 × 256 | 15.7606 | 0.1894 | 0.2334 | 121.5447 |
| Titok-bl128 | 256 × 256 | 16.3017 | 0.2373 | 0.1808 | 93.1151 |
| Titok-sl256 | 256 × 256 | 17.1632 | 0.3091 | 0.1413 | 69.7221 |
| OpenMagViT | 256 × 256 | 17.0070 | 0.3278 | 0.1406 | 70.6742 |
| LlamaGen ds8 | 256 × 256 | 19.1859 | 0.5144 | 0.0852 | 42.8318 |
| BSQ-ViT | 256 × 256 | **22.4836** | **0.7487** | **0.0496** | **28.5461** |
| VAR-256 | 256 × 256 | 17.8223 | 0.3960 | 0.1207 | 53.6363 |
| Janus Pro 1B/7B | 384 × 384 | 20.4726 | 0.4908 | 0.1174 | 59.6669 |
| Chameleon | 512 × 512 | 20.5686 | 0.5089 | 0.1182 | 67.0157 |
| LlamaGen ds16 | 512 × 512 | 21.8902 | 0.5903 | 0.0972 | 51.3484 |
| LlamaGen ds16 T2I | 512 × 512 | **21.9694** | 0.6029 | **0.0947** | **51.0251** |
| VAR-512 | 512 × 512 | 21.5492 | **0.6537** | 0.1000 | 57.3488 |
| Infinity-d32 | 1024 × 1024 | 29.1329 | 0.9052 | 0.0269 | 21.0871 |
| Infinity-d64 | 1024 × 1024 | 31.1374 | 0.9396 | 0.0188 | 16.7424 |
| SD3.5L | 1024 × 1024 | 37.4365 | 0.9774 | 0.0028 | 4.5717 |
| FLUX.1-dev | 1024 × 1024 | **41.5280** | **0.9908** | **0.0011** | **2.1202** |
| GPT-4o | 1024 × 1024 | 15.8109 | 0.3000 | 0.2156 | 75.8916 |

### 4.2 TASK 2: DETAIL PRESERVATION

In this task, we evaluate the ability of each VT to retain high-frequency visual information using a dataset of patterned and texture-rich images. As shown in Table 2, continuous VAEs again lead in all metrics. Among discrete VTs, Infinity-d64 demonstrate the strongest performance, suggesting that residual spherical normalization are effective for preserving detailed structures. We further visualize qualitative results in Figure 5. While continuous models such as SD3.5L, and FLUX.1 closely resemble the original, many VTs introduce visible distortions, such as blurred lines, broken shapes, and color bleeding. These results highlight the limitations of current discrete tokenization schemes in retaining local textures and structural integrity.

### 4.3 TASK 3: TEXT PRESERVATION

This task evaluates how well VTs preserve text content in images – a crucial capability for OCR, document understanding, and multimodal reasoning. We consider three increasingly challenging

Table 3: Evaluation of task 3 across three different settings.

| Method | Image Size | Movie Posters (Easy) | | | | | | ArXiv Abstracts (Hard) | | | | | | Multilingual (Non-Latin) | | | | | | | |
|---|---|---|---|---|---|---|---|---|---|---|---|---|---|---|---|---|---|---|---|---|---|
| | | PSNR↑ | SSIM↑ | LPIPS↓ | FID↓ | CER↓ | WER↓ | PSNR↑ | SSIM↑ | LPIPS↓ | FID↓ | CER↓ | WER↓ | PSNR↑ | SSIM↑ | LPIPS↓ | FID↓ | CER↓ (Chinese) | CER↓ (Hindi) | CER↓ (Japanese) | CER↓ (Korean) |
| FlowMo Lo | 256 × 256 | 24.1178 | 0.7471 | 0.0645 | 55.0685 | 0.6458 | 3.7022 | 14.4979 | 0.5961 | 0.0703 | 9.1933 | 0.8337 | 6.0968 | 13.3470 | 0.5694 | 0.1187 | 23.0079 | 1.2564 | 1.0397 | 1.3614 | 1.2823 |
| FlowMo Hi | 256 × 256 | 27.4186 | 0.8403 | 0.0380 | 40.7009 | 0.4169 | 2.5639 | 15.8510 | 0.7262 | 0.0493 | 6.6921 | 0.7046 | 5.3013 | 14.4191 | 0.6932 | 0.0896 | 11.8131 | 1.1279 | 0.9374 | 1.0281 | 0.9981 |
| MaskBiT 16bit | 256 × 256 | 21.0962 | 0.6748 | 0.0992 | 67.3479 | 0.7315 | 4.2234 | 14.3909 | 0.4616 | 0.1945 | 27.9940 | 1.3801 | 9.2146 | 13.2634 | 0.4157 | 0.3184 | 42.6548 | 3.1298 | 1.1953 | 2.0821 | 2.1311 |
| MaskBiT 18bit | 256 × 256 | 21.2932 | 0.6876 | 0.0928 | 64.0766 | 0.7910 | 4.8894 | 14.7015 | 0.4978 | 0.2436 | 26.2086 | 1.2828 | 9.2956 | 13.4963 | 0.4450 | 0.3450 | 59.1847 | 3.4889 | 1.1867 | 1.9445 | 2.3713 |
| Titok-l32 | 256 × 256 | 15.5164 | 0.4595 | 0.2616 | 124.7412 | 1.2254 | 7.8547 | 12.2423 | 0.2390 | 0.2513 | 60.0889 | 1.1178 | 8.5836 | 11.6064 | 0.2440 | 0.3557 | 78.3890 | 3.0154 | 1.1907 | 1.9135 | 1.7847 |
| Titok-b64 | 256 × 256 | 16.9423 | 0.5273 | 0.1939 | 101.7892 | 1.0189 | 6.7070 | 12.9500 | 0.3068 | 0.2470 | 49.4931 | 1.2414 | 10.0394 | 11.7860 | 0.2557 | 0.3163 | 82.9833 | 3.5650 | 1.4384 | 2.0044 | 2.3986 |
| Titok-s128 | 256 × 256 | 17.0506 | 0.5427 | 0.1598 | 88.7466 | 1.2478 | 6.2260 | 12.9276 | 0.3297 | 0.2691 | 56.6373 | 1.4054 | 8.6644 | 11.7860 | 0.2710 | 0.3131 | 64.4152 | 2.7573 | 1.1329 | 1.6614 | 1.9083 |
| Titok-bl64 | 256 × 256 | 18.2597 | 0.5963 | 0.1584 | 95.2880 | 1.1421 | 6.7044 | 13.4414 | 0.4535 | 0.1398 | 26.4375 | 1.2385 | 9.9727 | 12.1882 | 0.3908 | 0.1801 | 38.8779 | 3.6473 | 1.2105 | 2.8315 | 2.5622 |
| Titok-bl128 | 256 × 256 | 19.8102 | 0.6502 | 0.1159 | 77.7980 | 1.5956 | 5.0851 | 13.5015 | 0.4810 | 0.1235 | 24.1506 | 1.0432 | 7.6730 | 12.5783 | 0.4306 | 0.2012 | 36.9834 | 2.3695 | 0.9719 | 2.1597 | 1.8533 |
| Titok-sl256 | 256 × 256 | 21.4568 | 0.7151 | 0.0841 | 61.0830 | 0.7850 | 4.5561 | 13.4954 | 0.4996 | 0.1122 | 19.0590 | 1.2131 | 8.2139 | 12.7078 | 0.4669 | 0.1589 | 26.0457 | 2.2980 | 1.2338 | 1.9775 | 1.7088 |
| OpenMagViT | 256 × 256 | 22.5337 | 0.7440 | 0.0741 | 52.7153 | 0.7385 | 4.2419 | 13.6459 | 0.5472 | 0.0900 | 19.0211 | 1.0400 | 7.7046 | 12.3496 | 0.4874 | 0.1498 | 25.5786 | 2.5913 | 1.1622 | 2.0300 | 1.9826 |
| LlamaGen ds8 | 256 × 256 | 26.0990 | 0.8288 | 0.0371 | 31.4534 | 0.5066 | 2.9620 | 16.0494 | 0.7003 | 0.0648 | 25.1991 | 0.9769 | 7.1170 | 14.5250 | 0.6513 | 0.1024 | 19.2362 | 1.1408 | 1.0250 | 1.0505 | 1.1342 |
| BSQ-ViT | 256 × 256 | **30.7305** | **0.9135** | **0.0172** | **25.6235** | 1.1102 | 0.5763 | **20.3628** | **0.8910** | **0.0270** | **4.7719** | 0.8910 | 0.3234 | **17.9849** | **0.8544** | 0.0582 | 10.2721 | 0.7858 | 0.4722 | 0.3827 | 0.5919 |
| VAR-256 | 256 × 256 | 23.6215 | 0.7349 | 0.0602 | 50.2971 | 0.6391 | 3.7266 | 15.2481 | 0.6799 | 0.0569 | 23.1849 | 1.0170 | 7.3612 | 13.1452 | 0.5672 | 0.1165 | 19.0211 | 2.3378 | 1.1118 | 1.8972 | 1.5754 |
| Janus Pro 1B/7B | 384 × 384 | 25.2793 | 0.8203 | 0.0545 | 45.4080 | 0.6628 | 3.7549 | 17.0286 | 0.6535 | 0.1003 | 38.3604 | 1.0645 | 7.8280 | 15.7220 | 0.6223 | 0.1367 | 53.0056 | 1.5413 | 1.1578 | 1.3463 | 1.3333 |
| Chameleon | 512 × 512 | 27.1939 | 0.8358 | 0.0430 | 39.8786 | 0.5782 | 3.1359 | 18.1545 | 0.7503 | 0.0588 | 8.1713 | 0.7158 | 5.1867 | 15.3719 | 0.6370 | 0.0977 | **15.5254** | 1.9706 | 0.9171 | 1.5444 | 1.4047 |
| LlamaGen ds16 | 512 × 512 | 27.3778 | **0.8717** | 0.0440 | 40.9660 | 0.4914 | 2.8774 | 19.0951 | 0.7601 | 0.0601 | 18.1659 | 0.9527 | 6.8741 | 17.2316 | 0.7068 | 0.1141 | 38.2967 | 1.2307 | 1.0938 | 1.0343 | 1.1108 |
| LlamaGen ds16 T2I | 512 × 512 | **27.4929** | 0.8697 | **0.0411** | **38.2986** | 0.4842 | 2.5741 | 18.8911 | 0.7735 | 0.0567 | 20.8214 | 0.9010 | 6.1850 | 17.2267 | 0.7400 | **0.0869** | 45.0646 | 0.9993 | 1.0305 | 0.8139 | 0.9775 |
| VAR-512 | 512 × 512 | 26.4574 | 0.8331 | 0.0486 | 46.6674 | **0.1949** | **1.1867** | 21.0023 | **0.8561** | 0.0553 | 5.3145 | 0.1341 | 0.9829 | **17.7790** | **0.7687** | 0.1012 | 22.1121 | 0.8398 | 0.6873 | 0.6552 | 0.7067 |
| Infinity-d32 | 1024 × 1024 | 37.7533 | 0.9669 | 0.0063 | 10.5845 | 0.0790 | 0.4371 | 31.7028 | 0.9896 | 0.0049 | 0.7982 | 0.0017 | 0.0162 | 30.2736 | 0.9892 | 0.0086 | 2.2108 | 0.1758 | 0.0352 | 0.0661 | 0.0924 |
| Infinity-d64 | 1024 × 1024 | 39.3334 | 0.9774 | 0.0048 | 8.6894 | 0.0439 | 0.2978 | 35.0924 | 0.9952 | 0.0026 | 0.7426 | 0.0019 | 0.0165 | 34.5168 | 0.9959 | 0.0035 | 1.1291 | 0.1372 | 0.0604 | 0.0617 | 0.0582 |
| SD3.5L | 1024 × 1024 | 38.9058 | 0.9650 | 0.0011 | 1.6215 | **0.0002** | **0.0000** | 40.1536 | 0.9988 | 0.0005 | 0.1331 | 0.0005 | 0.0010 | 39.6703 | 0.9989 | 0.0011 | 0.4205 | **0.0973** | 0.0310 | 0.0107 | 0.0545 |
| FLUX.1-dev | 1024 × 1024 | **44.3863** | **0.9862** | **0.0005** | **0.8993** | 0.0521 | 0.4872 | **52.1000** | **0.9999** | **0.0001** | **0.0256** | 0.0018 | 0.0112 | **51.9243** | **0.9999** | **0.0001** | 0.0457 | 0.1220 | 0.0272 | 0.0614 | 0.0506 |
| GPT-4o | 1024 × 1024 | 17.7677 | 0.6006 | 0.1388 | 63.7351 | 0.0180 | 0.0836 | 11.3299 | 0.1516 | 0.1929 | 32.5382 | 0.4928 | 3.5292 | 10.4376 | 0.1761 | 0.2291 | 20.0185 | 0.8096 | 0.6414 | 0.7219 | 0.6478 |

scenarios: (1) Movie Posters with clean, short English text; (2) ArXiv Abstracts containing dense, long-form academic writing; and (3) Multilingual Text rendered in non-Latin scripts (Chinese, Hindi, Japanese, Korean). We use Character Error Rate (CER) and Word Error Rate (WER), based on OCR outputs from the Gemma 3 model, to quantify text preservation. As shown in Table 3 and Figure 5, continuous VAEs such as FLUX.1-dev and SD3.5L consistently outperform discrete tokenizers across conditions. Notably, many VTs fail to reconstruct slogans or titles in the Movie Poster setting, and perform worse on academic layouts or multilingual scripts. These results highlight the limitations of discrete tokenization methods in preserving symbolic fidelity, particularly under complex formatting and linguistic diversity. Additional experiments with alternative OCR backends are reported in Appendix G, confirming these findings robust to the choice of OCR system.

## 5 DISCUSSION

**Limitation of Existing VTs.** Discrete VTs fall significantly behind continuous VAEs in reconstruction fidelity, detail preservation, and symbolic accuracy, particularly for high-resolution, variable-size, and multilingual inputs. Many models are constrained by fixed input sizes and struggle to retain semantic structure in complex settings. Moreover, the absence of a strong, open-source VT limits progress in large-scale AR models. While LLMs continue to scale and improve, their performance is increasingly bottlenecked by the quality of visual tokenization. There is an urgent need for a resolution-flexible, semantically robust, and reusable VT that can keep pace with the capabilities of modern LLMs. We hope that VTBench can help accelerate research in this direction by providing a unified framework for diagnosis, comparison, and future development.

**Insights from GPT-4o Image Generation.** We present additional experiments comparing GPT-4o's image generation quality with that of diffusion models in Appendix I. The results suggest that GPT-4o may employ an autoregressive generation mechanism, aligning with prior research hypotheses (Yan et al., 2025). GPT-4o appears to inherit the knowledge and reasoning capabilities of LLMs while achieving strong visual synthesis quality, indicating its potential as a unified multimodal model. Although the exact architecture of GPT-4o's VT remains undisclosed, the high quality of its generated images implies a highly capable VT design. Based on qualitative analysis, we hypothesize that GPT-4o may use a residual next-scale VAE (RVAE) (Tian et al., 2024; Han et al., 2024) or a diffusion-based encoder-decoder (Yan et al., 2025). We consider the former more likely, as residual tokenization naturally aligns with the input-output format of LLMs and supports AR generation within a unified framework. This structure would allow the model to seamlessly integrate image understanding and generation, facilitating effective multimodal learning.

## 6 CONCLUSION

In this paper, we introduce VTBench, a comprehensive benchmark designed to systematically evaluate the performance of VTs in AR image generation pipelines. Our benchmark spans three tasks: Image Reconstruction, Detail Preservation, and Text Preservation, covering diverse scenarios. Through extensive experiments on a wide range of SOTA VTs, we uncover that discrete VTs fall substantially behind continuous VAEs, particularly in reconstruction fidelity, symbolic accuracy, and spatial consistency. These limitations are further magnified under complex visual conditions. By providing a unified evaluation framework, VTBench aims to bridge the gap between visual and language modalities, encourage the development of strong open-source VTs, and support the broader goal of building unified, multimodal generative algorithms.

ETHICS STATEMENT

All authors have read and will adhere to the ICLR Code of Ethics. Our study uses publicly available datasets (ImageNet-1k val, DIV2K test) and synthetically generated images created via programmatic prompts; no human subjects or personal data are involved. For arXiv abstracts, we reference public identifiers and provide rendering code rather than redistributing restricted text. We took care to avoid harmful or offensive content in the synthetic sets and to document fonts and layouts for multilingual renders to mitigate script-specific bias. Third-party models and assets are used under their respective licenses and appropriately credited. There are no conflicts of interest or external sponsorship influencing this work. We will release the dataset variants produced for VTBench and the full codebase to support transparency and community scrutiny.

REPRODUCIBILITY STATEMENT

We provide an anonymized repository with complete evaluation code, configuration files, and instructions to reproduce all figures and tables. The repository includes exact preprocessing, metric implementations (PSNR, SSIM, LPIPS, FID, CER, WER), fixed splits for each task, prompts and rendering pipelines for synthetic data, and pointers or scripts to obtain any licensed sources. Software and hardware details and versioning are documented, and random seeds are fixed where applicable; sources of nondeterminism (e.g., external image APIs) are noted with our selection protocol. Upon publication, we will open-source both the VTBench dataset and the code to facilitate independent verification and extension.

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

## A   THE USE OF LLMs

Large Language Models (LLMs) were used solely to aid and polish the writing of this paper, improving grammar, clarity, and readability. They were not involved in research ideation, experimental design, implementation, or analysis. The authors bear full responsibility for all content presented.

## B   EXTENDED BACKGROUND ON VISUAL TOKENIZERS

While the main text provides a brief overview of VTs, this section offers a more detailed explanation of representative visual tokenizers, including both continuous and discrete variants. We present formal definitions and architectural insights for each method, laying the groundwork for understanding their differences in performance and suitability for autoregressive image generation.

**Continuous VAE.** In diffusion models, a common approach is to use a continuous Variational Autoencoder (VAE) (Rombach et al., 2022; Peebles & Xie, 2023) as a feature compressor as shown in Figure 2(a). These VAEs encode the input image into a continuous latent space, typically using convolutional encoders and decoders, enabling high-quality image reconstructions. Formally, let $x \in \mathbb{R}^{H \times W \times 3}$ be the input image, where $H$ and $W$ denote the image's height and width, respectively. The encoder $E$ maps $x$ to a latent representation $z = E(x) \in \mathbb{R}^{h \times w \times d}$, where $h$ and $w$ are the spatial dimensions of the latent space (typically downsampled from $H$ and $W$), and $d$ is the latent channel dimension. The decoder $D$ reconstructs the image as $\hat{x} = D(z)$. However, AR models require inputs in the form of discrete token sequences, motivating the development of discrete visual tokenizers.

**VQ-VAE.** Vector Quantized Variational Autoencoder (VQ-VAE) (van den Oord et al., 2017) is one of the earliest and most widely adopted discrete visual tokenizers. It encodes an image into a latent feature map and then quantizes each spatial location to the closest vector in a learned codebook. The resulting discrete indices form a token grid that can be used in AR models. Given an input image $x \in \mathbb{R}^{H \times W \times 3}$, the encoder $E$ produces a latent map $z = E(x) \in \mathbb{R}^{h \times w \times d}$. Each latent vector $z_{i,j}$ is replaced by its nearest neighbor in the codebook $\mathcal{C} = \{c_1, c_2, \ldots, c_K\} \subset \mathbb{R}^d$, $q_{i,j} = \arg\min_k ||z_{i,j} - c_k||_2^2$, $\hat{z}_{i,j} = c_{q_{i,j}}$. The quantized latent $\hat{z}$ is then decoded via $D$ to reconstruct the image $\hat{x} = D(\hat{z})$. While VQ-VAE enables discrete tokenization suitable for AR models, it suffers from limited expressivity, codebook collapse, and scalability issues when handling high-resolution or diverse image content.

**LFQ-VAE.** Lookup-Free Quantization (LFQ) (Yu et al., 2024a) eliminates the nearest-neighbor lookup and codebook structure by directly projecting latent features into a binary latent space using a learnable mapping. The encoder $E$ produces $z = E(x) \in \mathbb{R}^{h \times w \times d}$. Each element $z_{i,j,k}$ is quantized using a sign function: $\hat{z}_{i,j,k} = \text{sign}(z_{i,j,k}) \in \{-1, +1\}$. The binary representation can be interpreted as a discrete token mask $m$, where each binary vector at location $(i, j)$ is mapped to an integer token using $m_{i,j} = \sum_{k=1}^{d} 2^{k-1} \mathbb{1}_{z_{i,j,k} > 0}$.

**BSQ-VAE.** Binary Spherical Quantization (BSQ) (Zhao et al., 2024) builds upon LFQ by incorporating $\ell_2$ normalization before quantization: $\hat{z}_{i,j,k} = \frac{1}{|\sqrt{d}|} \text{sign}(\frac{z_{i,j,k}}{|z_{i,j}|})$. This constrains the representation to a unit hypersphere, effectively reducing quantization error and leading to smoother latent spaces. BSQ allows for finer-grained tokenization while maintaining efficient binary encoding, which is beneficial for downstream tasks sensitive to visual detail.

**Residual Next-Scale VAE** introduces hierarchical tokenization by progressively quantizing residual information across multiple spatial scales (Tian et al., 2024). Let $s \in \{1, 2, \ldots, n\}$ denote the scale index, where $n$ is the total number of scales. Each scale $s$ corresponds to a predefined spatial resolution $(h_s, w_s)$, with $h_s \leq h$ and $w_s \leq w$. During encoding, a sequence of quantized tokens $\hat{z} = \{\hat{z}^{(1)}, \hat{z}^{(2)}, \ldots, \hat{z}^{(n)}\}$ is computed as follows:

$$\begin{cases} z = r^{(0)} = E(x) \in \mathbb{R}^{h \times w \times d} \\ \hat{z}^{(s)} = \mathcal{Q}\left(\downarrow_{(h_s, w_s)} \left(r^{(s-1)}\right)\right), \ r^{(s)} = z - \uparrow_{(h,w)} \left(\mathcal{D}\left(\hat{z}^{(s)}\right)\right), \quad s = 1, \ldots, n \end{cases} \tag{1}$$

Here, $\mathcal{Q}$ and $\mathcal{D}$ represent the quantization and dequantization functions, respectively, which may be implemented using VQ, LFQ, BSQ, or similar methods. The operators $\downarrow_{(h_s, w_s)} (\cdot)$ and $\uparrow_{(h,w)} (\cdot)$ denote downsampling to and upsampling from the specified resolution. During decoding, the final image is reconstructed by aggregating and decoding all quantized components:

$$\hat{x} = D\left(\sum_{s=1}^{n} \uparrow_{(h,w)} \left(\mathcal{D}(\hat{z}^{(s)})\right)\right) \tag{2}$$

This coarse-to-fine scheme preserves spatial structure and enables the model to capture both global context and fine-grained details, making it effective for high-resolution image reconstruction.

## C  IMAGE SYNTHESIS

In this section, we detail the procedure for synthesizing high-resolution and detailed images.

Generally, in our experiments, we utilized the `openai.images.edit` API with the `gpt-image-1` model. For GPT-4o, we retried the API call until a successful image generation was achieved. Fortunately, we encountered no issues aside from occasional connection or timeout errors, which were typically resolved within one or two attempts. As image generation with GPT-4o is nondeterministic and the API does not allow setting the temperature or a random seed, we consistently used the first successfully generated image for evaluation. Notably, we observed that the images generated from the same input were generally similar in appearance.

**High Resolution Images.** To generate diverse scene descriptions, we use the prompt: *"I want to generate high-resolution images with a size of $1024 \times 1024$. Please directly create 10 different fictional scenes for me and output them in JSONL format."* This prompt is repeated 10 times, resulting in 100 unique scene descriptions. Each description is then passed to GPT-4o's image generation API to produce 100 corresponding images, each with a resolution of $1024 \times 1024$.

**Detailed Images.** Following a similar approach to high-resolution image synthesis, we employed the following prompt: *"I want to design a richly detailed dataset, featuring abundant lines, color blocks, textures, or various forms of high-frequency information. The scenes do not need to be realistic. I plan to use image generation algorithms to produce these images. Please help me design 5 different prompts and output them in JSONL format."* This prompt is issued 10 times to generate a total of 100 unique scene descriptions. Each description is then used with GPT-4o's image generation API to synthesize 100 corresponding images at a resolution of $1024 \times 1024$.

**Movie Poster Synthesis** To synthesize movie posters, we first generate fictional movie titles and subtitles using the prompt: *"Please generate a fictional movie titles of different types, along with subtitles, and output them in JSONL format."* These synthetic titles and subtitles are then used in a second prompt to create the corresponding posters: *'Generate a 1024×1024 movie poster for the film titled (movie title), incorporating the text: (subtitle). The title and slogan should blend naturally into the poster design.* This process is repeated 100 times to produce 100 unique movie posters.

## D  DEFINITION OF VISUAL DETAILS

The goal of Task 2 is to evaluate the ability of VTs to retain image details. In this section, we define what is meant by "details" in the context of Task 2.

***What Is "Detail"?*** In the context of VTBench, we define "visual detail" as high-frequency visual information that is essential for preserving the local semantic integrity of an image. Concretely,

we categorize details into the following visual elements: (1) Fine textures (e.g., fabric weave, hair strands, foliage patterns); and (2)Sharp edges and contours (e.g., architectural lines, object boundaries). These are typically spatially localized, semantically significant, and vulnerable to information loss during quantization. Unlike global semantics (e.g., category labels), these details are often non-redundant and not easily recoverable once degraded, especially in AR generation pipelines where token quality directly determines synthesis fidelity.

***Why This Definition Matters for Tokenization?*** Discrete VT compress image features into finite token vocabularies. During this compression, high-frequency information is the first to be lost, particularly under quantization or low expressiveness of token representations. This directly impacts downstream generation quality by perceptual degradation (textures appear smoothed, edges blurred, and small features missed) Therefore, evaluating a VT's ability to retain such localized, semantically relevant details is essential for both perceptual quality and task-level accuracy in multimodal applications.

Unlike global semantic content, such as scene category or layout, these local features are often non-redundant and highly sensitive to quantization errors. Preserving them is essential for perceptual quality and for downstream tasks that rely on accurate image content.

To make this definition operational and measurable, we designed Task 2: Detail Preservation specifically to target and evaluate a model's ability to retain such features. To validate that Task 2 indeed focuses on detail-rich content, we computed three standard image analysis metrics across all tasks as show in Table 4:

- **High-Frequency Energy (%):** This measures the proportion of image energy in high-frequency components (computed via FFT), indicating the global presence of textures and patterns.
- **Laplacian Variance:** This quantifies the overall sharpness of an image by measuring the variance of the Laplacian response, a standard proxy for edge clarity and detail strength.
- **Edge Density (%):** This captures the spatial density of edges in an image, revealing how concentrated and frequent fine structures are within a scene.

Table 4: Verification that Task 2 images contain richer details.

| Task | Subtask | HighFreq Energy(%) ↑ | Laplacian Variance ↑ | Edge Density(%) ↑ |
|------|---------|----------------------|----------------------|-------------------|
| Task 1 | Imagenet | 40.7911 | 2735.9673 | 11.0883 |
| | High Resolution | 40.3702 | 940.7714 | 5.0195 |
| | Varying Resolution | 34.1957 | 1188.534 | 9.3666 |
| Task 2 | **Detail Preservation** | **48.931** | **12852.3598** | **23.6103** |
| Task 3 | Movie Posters | 44.2046 | 1288.8568 | 2.3493 |
| | Arxiv Abstracts | 39.6323 | 9857.1757 | 9.1615 |
| | Multilingual (Chinese) | 31.5059 | 8212.4345 | 10.1427 |
| | Multilingual (Hindi) | 31.7221 | 8087.6291 | 9.0816 |
| | Multilingual (Japanese) | 32.3393 | 8691.5516 | 9.6565 |
| | Multilingual (Korean) | 29.983 | 6847.8147 | 9.5393 |

As shown in Table 4, among all benchmark tasks, Task 2 exhibits the highest values across all three indicators. Specifically, it has a High-Frequency Energy of 48.93%, a Laplacian Variance of 12,852.36, and an Edge Density of 23.61%. These values far exceed those of Task 1 (ImageNet-Val), which show only 40.79%, 2735.97, and 11.09%, respectively, as well as all variants of Task 3. For example, even the ArXiv Abstracts test set, which contains visually complex scientific documents, reaches only 39.63% High-Frequency Energy and 9.16% Edge Density.

This quantitative evidence demonstrates that Task 2 is constructed to challenge models on their ability to preserve fine-grained, localized, and perceptually significant visual features. By combining these measurements with perceptual similarity metrics such as LPIPS in evaluation, we ensure a rigorous and reproducible assessment of detail preservation quality.

# E    RESOLUTION LIMITATIONS OF DISCRETE VTS

Although discrete visual tokenizers (VTs) and continuous variational autoencoders (VAEs) share a similar high-level encoder–decoder paradigm, their behavior diverges substantially when handling arbitrary image resolutions. In practice, current discrete VTs exhibit several architectural and training limitations that hinder their ability to generalize across resolutions, in contrast to the fully convolutional structure of continuous VAEs.

**Input Size Constraints.**   Many discrete VTs, particularly those based on vector quantization (e.g., VQ-VAE, LFQ), require fixed input sizes during both training and inference. This constraint arises from codebook usage, patchification schemes, and decoder assumptions (e.g., flattening 2D grids into token sequences of fixed length). For example, models such as Janus support only predefined square resolutions (e.g., $256 \times 256$, $384 \times 384$, or $512 \times 512$). Processing unseen resolutions often results in runtime errors or corrupted outputs, as illustrated in Figure 4.

**Decoder Design Dependency.**   The decoders of discrete VTs typically assume a token grid shape to map sequences back into spatial layouts. This coupling binds input resolution to the decoder architecture. In contrast, convolution-based VAEs, such as those employed in diffusion models, naturally support arbitrary resolutions because of their fully convolutional encoder–decoder design.

**Token Flattening and Positional Encoding.**   Autoregressive models built on discrete tokens usually flatten spatial grids into one-dimensional sequences. This flattening exacerbates resolution sensitivity: token grid dimensions change with input size, but positional embeddings and transformer blocks are rarely designed to handle such variation. Consequently, discrete VTs exhibit reduced robustness when applied to arbitrary resolutions.

In summary, while quantization itself is not inherently resolution-sensitive, current implementations of discrete VTs are limited in practice by input size assumptions, decoder rigidity, and positional encoding designs. These architectural constraints make discrete VTs significantly less robust to varying resolutions compared to continuous VAEs.

# F    TEXT QUALITY EVALUATION

Evaluating the ability of a VT to preserve text content is critical for many downstream applications such as OCR, document generation/understanding, and multimodal reasoning. In VTBench, we quantify this ability by applying Optical Character Recognition (OCR) to both original and reconstructed images, and computing standard text similarity metrics based on the OCR outputs.

**Optical Character Recognition (OCR)** We use a SOTA multimodal model, Gemma 3 (Kamath et al., 2025), to extract textual content from both the original image $x$ and the reconstructed image $\hat{x}$. Let $T_{\text{orig}}$ and $T_{\text{recon}}$ denote the sequences obtained from $x$ and $\hat{x}$, respectively. All evaluations are performed under the same OCR configuration to ensure consistency.

**Character Error Rate (CER)** The Character Error Rate measures the normalized edit distance between two character sequences. Given the ground-truth sequence $T_{\text{orig}}$ and the predicted sequence $T_{\text{recon}}$, CER is computed as:

$$\text{CER} = \frac{D_{\text{char}}(T_{\text{orig}}, T_{\text{recon}})}{|T_{\text{orig}}|} \tag{3}$$

where $D_{\text{char}}(\cdot, \cdot)$ denotes the Levenshtein distance (i.e., the minimum number of insertions, deletions, and substitutions needed to convert one character sequence to another), and $|T_{\text{orig}}|$ is the number of characters in the ground-truth sequence.

**Word Error Rate (WER).** Word Error Rate is similar to CER, but computed at the word level. Let $W_{\text{orig}}$ and $W_{\text{recon}}$ be the sequences of words extracted from the original and reconstructed images, respectively. The WER is defined as:

$$\text{WER} = \frac{D_{\text{word}}(W_{\text{orig}}, W_{\text{recon}})}{|W_{\text{orig}}|} \tag{4}$$

where $D_{\text{word}}(\cdot, \cdot)$ is the Levenshtein distance between word sequences, and $|W_{\text{orig}}|$ is the number of words in the ground-truth text.

As with CER, WER can also exceed 1 in extreme failure cases, where the number of incorrect or missing words far surpasses the original word count. These metrics jointly capture both low-level character accuracy and higher-level semantic correctness. High CER or WER indicates that the visual tokenizer has failed to preserve text layout or visual clarity during reconstruction, especially in multilingual or high-density text scenarios.

## G  ROBUSTNESS OF TEXT PRESERVATION TO OCR CHOICE

Our text preservation evaluation in Task 3 relies on OCR outputs to compute character error rate (CER) and word error rate (WER). A natural concern is whether the results are sensitive to the choice of OCR system. To investigate this, we conducted a follow-up experiment with two additional OCR backends beyond Gemma 3: `Qwen2.5-VL-7B-Instruct` and `nanonets/Nanonets-OCR-s`. The CER/WER results across all three OCR systems are summarized in Table 5.

Table 5: Comparison of OCR sensitivity in text preservation evaluation. Results are consistent across Gemma 3, Qwen2.5-VL-7B-Instruct, and Nanonets-OCR-s.

| Method | Gemma 3 | | Qwen2.5-VL-7B-Instruct | | Nanonets-OCR-s | |
| --- | --- | --- | --- | --- | --- | --- |
| | CER | WER | CER | WER | CER | WER |
| FlowMo Lo | 0.6458 | 3.7022 | 0.5969 | 3.4291 | 0.5861 | 3.3744 |
| FlowMo Hi | 0.4169 | 2.5639 | 0.4148 | 2.3856 | 0.3987 | 2.2985 |
| MaskBiT 16bit | 0.7315 | 4.2234 | 0.7679 | 4.4453 | 1.0437 | 6.0299 |
| MaskBiT 18bit | 0.7910 | 4.8894 | 0.7599 | 4.3905 | 0.7330 | 4.2090 |
| Titok-l32 | 1.2254 | 7.8547 | 0.9549 | 5.5398 | 2.1851 | 10.1903 |
| Titok-b64 | 1.0189 | 6.7070 | 1.1570 | 5.5000 | 1.3111 | 7.5522 |
| Titok-s128 | 1.2478 | 6.2260 | 0.9538 | 8.1082 | 1.1665 | 6.7177 |
| Titok-bl64 | 1.1421 | 6.7044 | 1.1605 | 6.7313 | 1.9725 | 6.4577 |
| Titok-bl128 | 1.5956 | 5.0851 | 1.2116 | 7.8060 | 2.6343 | 15.3209 |
| Titok-sl256 | 0.7850 | 4.5561 | 0.7634 | 4.4179 | 0.7377 | 4.2376 |
| OpenMagViT | 0.7385 | 4.2419 | 0.6932 | 4.0100 | 0.8561 | 4.9428 |
| LlamaGen ds8 | 0.5066 | 2.9620 | 0.5471 | 3.1679 | 0.4913 | 2.8445 |
| BSQ-VIT | 0.1102 | 0.5763 | 0.1059 | 0.6480 | 0.0659 | 0.3495 |
| VAR-256 | 0.6391 | 3.7266 | 0.6243 | 3.6206 | 0.6130 | 3.5448 |
| Janus Pro 1B/7B | 0.6628 | 3.7549 | 0.6488 | 3.7920 | 1.0328 | 4.7400 |
| Chameleon | 0.5782 | 3.1359 | 0.5369 | 3.1295 | 0.7092 | 4.0871 |
| LlamaGen ds16 | 0.4914 | 2.8774 | 0.5112 | 2.9676 | 0.4704 | 2.6841 |
| LlamaGen ds16 T2I | 0.4842 | 2.5741 | 0.4868 | 2.7983 | 0.4470 | 2.5734 |
| VAR-512 | 0.1949 | 1.1867 | 0.1263 | 0.7472 | 0.1121 | 0.6281 |
| Infinity-d32 | 0.0790 | 0.4371 | 0.0098 | 0.0548 | 0.0026 | 0.0025 |
| Infinity-d64 | 0.0439 | 0.2978 | 0.0043 | 0.0237 | 0.0019 | 0.0000 |
| SD3.5L | 0.0002 | 0.0000 | 0.0004 | 0.0025 | 0.0045 | 0.0062 |
| FLUX.1-dev | 0.0521 | 0.4872 | 0.0004 | 0.0025 | 0.0013 | 0.0012 |
| GPT-4o | 0.0180 | 0.0836 | 0.0186 | 0.1034 | 0.0173 | 0.0784 |

Despite differences in absolute CER/WER values, the relative ranking of visual tokenizers remains stable across all three OCR systems. High-performing VTs such as BSQ-VIT, Infinity-d64, and FLUX.1-dev consistently outperform weaker tokenizers such as Titok variants or MaskBiT. These results confirm that the trends reported in Table 3 are robust to the choice of OCR backend, and our conclusions regarding text preservation are not artifacts of a particular OCR model.

| Original | InstructPix2Pix | Step1X-Edit | IC-Edit | GPT-4o | GPT-4o-OCR (Extract from original image) |
|---|---|---|---|---|---|

Figure 6: Comparison of image editing using the prompt *"Please recreate the exact same image without any alterations."* with the original image. The original images contain blurred text that is not recognizable to humans. The rightmost column displays the OCR-extracted text from the original image using GPT-4o using the prompt *"Directly output what is written in the image."*

## H  EXPERIMENTAL SETTINGS AND ENVIRONMENTS

**Experimental Settings.** We evaluate a range of SOTA VTs across the three core tasks in VT-Bench: Image Reconstruction, Detail Preservation, and Text Preservation. For each task, we use a fixed evaluation protocol to ensure consistency across models. All tokenizers are evaluated in an inference-only setting using publicly available pretrained weights. Where applicable, we follow each model's official preprocessing pipeline and input size specification (see Table 1). For text preservation evaluation, OCR is applied to both original and reconstructed images before computing CER and WER. We use the same set of 50,000 ImageNet validation images and 100 samples per text benchmark (movie posters, abstracts, multilingual) for all models.

**Environments.** All experiments are conducted using 4 NVIDIA A100 80GB GPUs with 251GB memory. Image preprocessing, reconstruction, and OCR evaluation are run on a standardized, reproducible pipeline. All results are collected on the same hardware and software stack to ensure fair comparisons.

## I  ADDITIONAL EXPERIMENTS OF GPT-4O IMAGE GENERATION

To further understand the GPT-4o's architecture and VT, we conducted a set of controlled experiments focused on its capabilities in font recognition and fine-grained text editing. These experiments are designed to isolate and probe the internal mechanisms behind GPT-4o's image generation and editing abilities, specifically evaluating the underlying VT's expressiveness and semantic alignment.

### I.1  IMAGE RECONSTRUCTION FROM LOW-QUALITY INPUTS

In this experiment, we provided GPT-4o with an intentionally degraded input image containing heavily distorted or unreadable text. Our prompt explicitly instructed the model to *recreate the image exactly without any alteration*. Interestingly, unlike diffusion-based models, which tend to replicate the visual noise and illegibility, GPT-4o instead produced an image where the text had been cleanly restored into readable, coherent sentences, as shown in Figure 6.

This behavior strongly deviates from the expected literal replication and instead suggests that GPT-4o engaged in a form of semantic hallucination. We hypothesize that GPT-4o first attempts to understand the image content through an internal representation, and in doing so, "fills in" the missing or noisy parts based on learned linguistic priors. To test this hypothesis, we queried GPT-4o to extract text from the original distorted image. Surprisingly, the model returned well-formed and highly plausible text, even though no readable characters are actually present in the image. Moreover, the

| Original | InstructPix2Pix | Step1X-Edit | IC-Edit | GPT-4o |
|---|---|---|---|---|

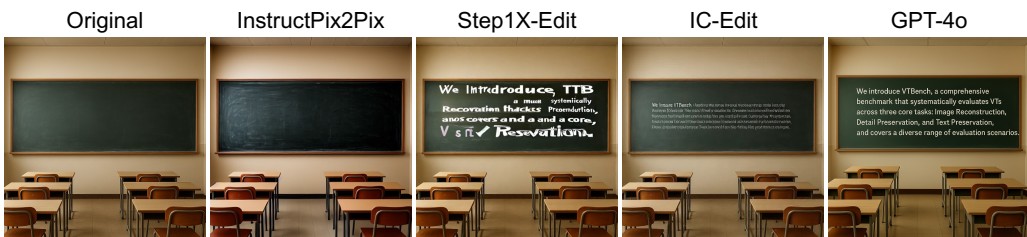

Prompt: Add text "We introduce VTBench, a comprehensive benchmark that systematically evaluates VTs across three core tasks: Image Reconstruction, Detail Preservation, and Text Preservation, and covers a diverse range of evaluation scenarios." to the blackboard.

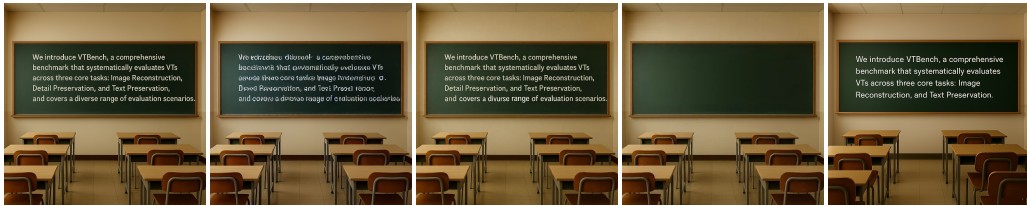

Prompt: Remove the text "and covers a diverse range of evaluation scenarios." from the blackboard.

Figure 7: Text editing comparison across models, showing both text insertion (top) and removal (bottom) on a blackboard.

extracted text closely matched the regenerated image, indicating a strong coupling between image understanding and generation.

These observations suggest that GPT-4o has inherited a strong capacity for image understanding. The model does not simply replicate noisy input, but instead attempts to semantically interpret and "restore" the image content, even when such interpretation involves hallucination. This behavior implies that GPT-4o's backbone is not a generic image generation model, but one that possesses robust multimodal reasoning capabilities. We hypothesize that this capability arises either from: (1) **A shared autoregressive backbone** pretrained or co-trained for both language and vision tasks, enabling semantic-level image understanding and generation, or (2) **A language-centric model** (e.g., GPT-4) that has been subsequently fine-tuned on multimodal tasks, thereby acquiring the ability to parse and reconstruct image content via linguistic priors.

In either case, the consistency between the model's interpretation and generation of text, despite degraded visual input, strongly supports the view that GPT-4o performs image generation not as a pixel-to-pixel transformation task, but as a language-informed semantic generation process.

### I.2 Text Addition and Deletion in Scene Context

To further explore GPT-4o's ability to understand and manipulate visual content, we conducted a series of experiments focused on localized text editing – inserting and removing words from realistic visual scenes, such as blackboards and signage. These tasks require not only accurate prompt-following, but also semantic understanding of scene layout, font style, and background context.

In the text addition experiments, GPT-4o was prompted to insert specific phrases into designated regions of an image, as shown in Figure 7. The results show remarkable spatial precision and stylistic coherence: the added text matched the surrounding content in font, size, alignment, and even lighting and shading. In the deletion experiments, GPT-4o is instructed to remove certain words. Rather than crudely masking out regions, the model naturally inpaint the background, reconstructing the texture and structure behind the deleted text with minimal artifacts.

These results reveal several important properties of GPT-4o's architecture:

- **Semantic-level image editing**: The model does not simply manipulate pixels, but appears to operate at a higher level of abstraction. It understands which parts of the image correspond to the textual prompt and modifies them while preserving global visual coherence.
- **Language-conditioned spatial reasoning**: The edits are guided by natural language, which suggests a strong alignment between textual and visual representations. This is indicative of a shared or tightly coupled multimodal representation space.

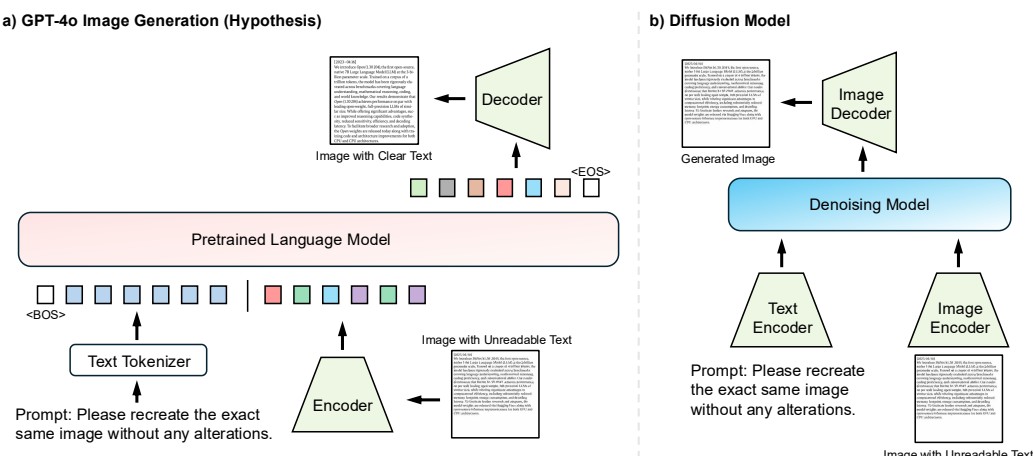

Figure 8: Comparison of two hypothesized architectures for image generation.

- **Unified generation and understanding backbone**: The precision and fidelity of the edits, especially the seamless integration of new content, suggest that the same underlying autoregressive model is responsible for both interpreting the image and generating the modified output. This unified design contrasts with traditional pipelines that separate perception and generation.

Together with the hallucination behavior observed in the font reconstruction experiment, these findings support the view that GPT-4o integrates image understanding and image generation into a single, coherent process. It reasons about images in a way that is language-centric, and its visual tokenizer must be capable of encoding not only fine-grained appearance but also semantic structure and spatial context.

## I.3 Autoregressive vs. Diffusion: Implications for GPT-4.

As summarized in Figure 8, we hypothesize that GPT-4o adopts a unified autoregressive architecture rather than a diffusion-based pipeline. This hypothesis is supported by the following observations:

- **Semantic hallucination through language priors:** GPT-4o's tendency to "restore" unreadable text into coherent language, despite prompts requesting exact replication suggests that its image understanding is conditioned by a language model prior. Such behavior is natural in an autoregressive framework where image and text tokens are processed in a shared sequence and jointly influenced by a pretrained language model.
- **Language-grounded editing and generation:** The model demonstrates precise text insertion and deletion aligned with natural language instructions, which requires not only generation capability but also deep semantic understanding of both visual context and instruction intent. This is hard to achieve in diffusion models, where the denoising model lacks linguistic grounding or token-level reasoning.
- **Modality reuse and efficiency:** The autoregressive formulation allows GPT-4o to reuse the same pretrained language model for both textual and visual reasoning, leveraging strong LLM priors for multimodal understanding. In contrast, diffusion models typically require separate encoders and decoders for each modality, making such deep integration more difficult.
- **Failure mode contrast:** Diffusion models tend to replicate noise patterns or preserve unreadability when asked to reconstruct degraded text, faithfully following low-level visual patterns. GPT-4o, on the other hand, produces semantically enriched outputs, even when that contradicts pixel-level fidelity, indicative of language-centric generation.

Together, these insights point toward GPT-4o employing an autoregressive backbone with a high-capacity visual tokenizer that supports semantic, instruction-driven generation. This structure allows GPT-4o to unify image understanding and image synthesis in a way that is tightly aligned with language modeling capabilities.

### I.4 GPT-4O'S VISUAL TOKENIZER

Despite the impressive capabilities exhibited by GPT-4o in both image understanding and generation, its VT remains proprietary and undisclosed. Nevertheless, its behavior provides strong indirect evidence of a highly expressive VT design: it preserves fine-grained textual structure, aligns visual and linguistic semantics, and supports high-resolution, editable image representations. Unlike many existing discrete tokenizers that struggle with text reconstruction and spatial fidelity, GPT-4o's VT appears to encode images in a way that is not only compact, but also semantically rich and compatible with autoregressive decoding.

To replicate such a tokenizer, future work must prioritize three core capabilities:

- **High-fidelity image reconstruction:** The tokenizer should retain sufficient visual detail to allow accurate reconstruction of the input image, ideally matching the performance of state-of-the-art continuous VAEs in terms of perceptual quality and structure preservation.
- **Resolution scalability:** It should generalize across varying image sizes and support high-resolution inputs without introducing artifacts or requiring fixed input dimensions, a common limitation of many existing discrete tokenizers.
- **Compatibility with downstream language models:** The output token sequences must be structurally and semantically aligned with the expectations of large language models (LLMs), enabling seamless integration into autoregressive multimodal frameworks for joint reasoning and generation.

Architectures such as residual next-scale VAEs, hierarchical quantization, or continuous tokenization may serve as promising directions. Given the lack of public access to GPT-4o's VT, benchmarks like VTBench offer a practical path forward: by systematically evaluating VT components in isolation, researchers can identify weaknesses, measure progress, and guide the development of open-source alternatives that approach or surpass GPT-4o's performance.

## J ADDITIONAL VISUALIZE QUALITATIVE RESULTS

To complement the quantitative evaluations presented in the main paper, we provide additional qualitative results that illustrate the reconstruction quality, text preservation, and detail retention of various visual tokenizers, from Figure 9 to 13.

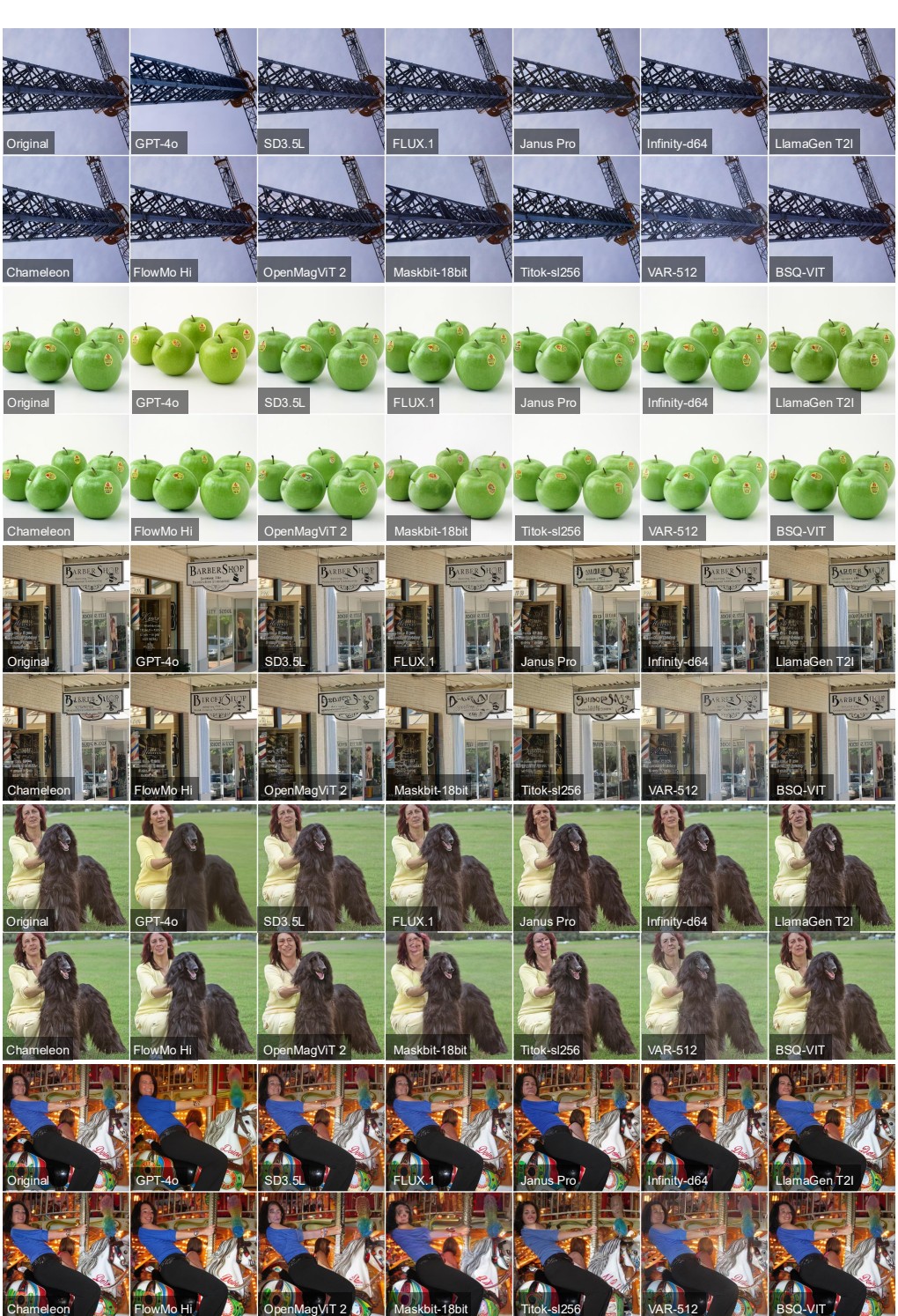

Figure 9: Additional visualize qualitative results for imagenet of task 1.

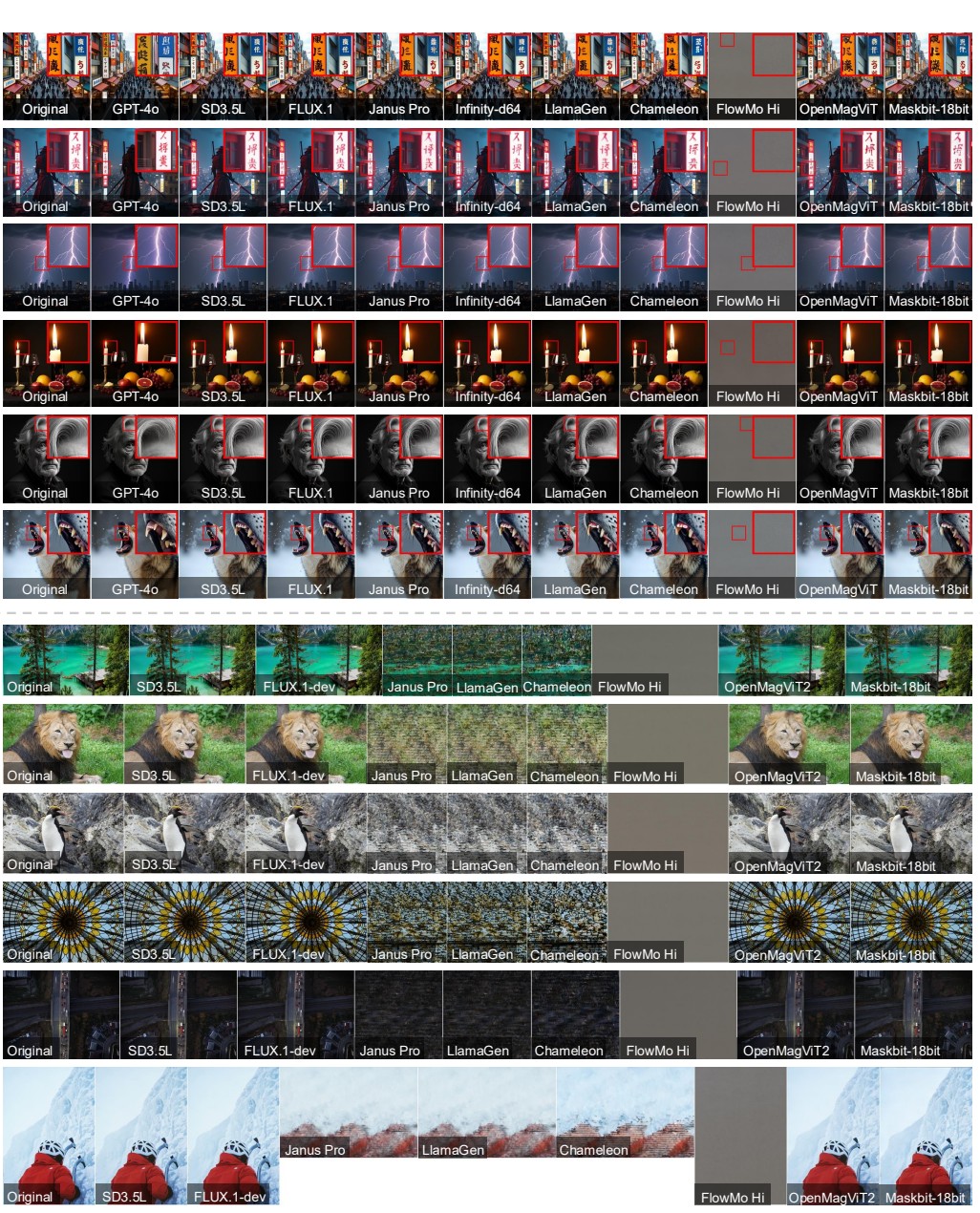

Figure 10: Visualized examples for high-resolution (top) and varying-resolution (bottom) of task 1.

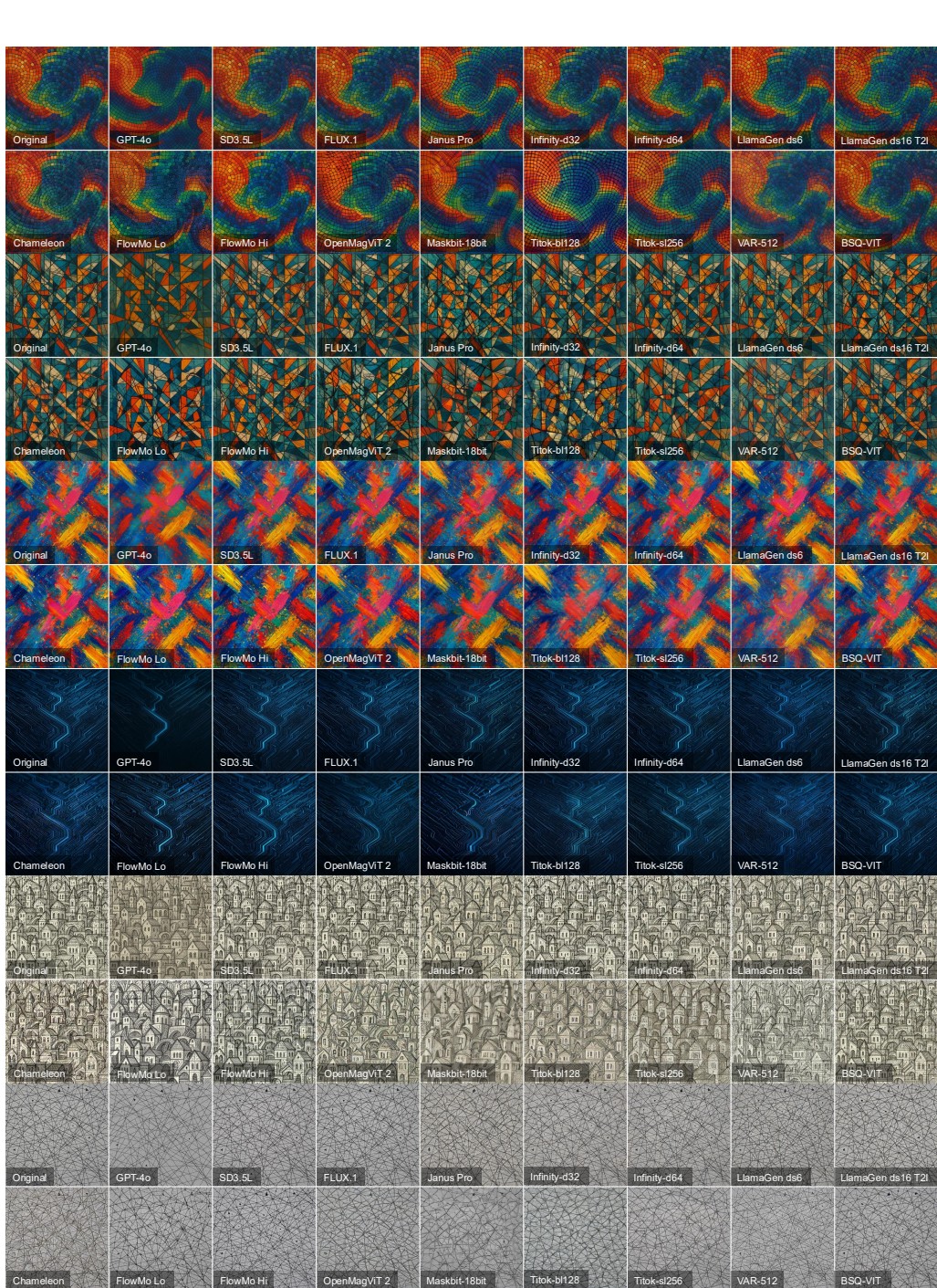

Figure 11: Additional visualize qualitative results for detail reservation (task 2).

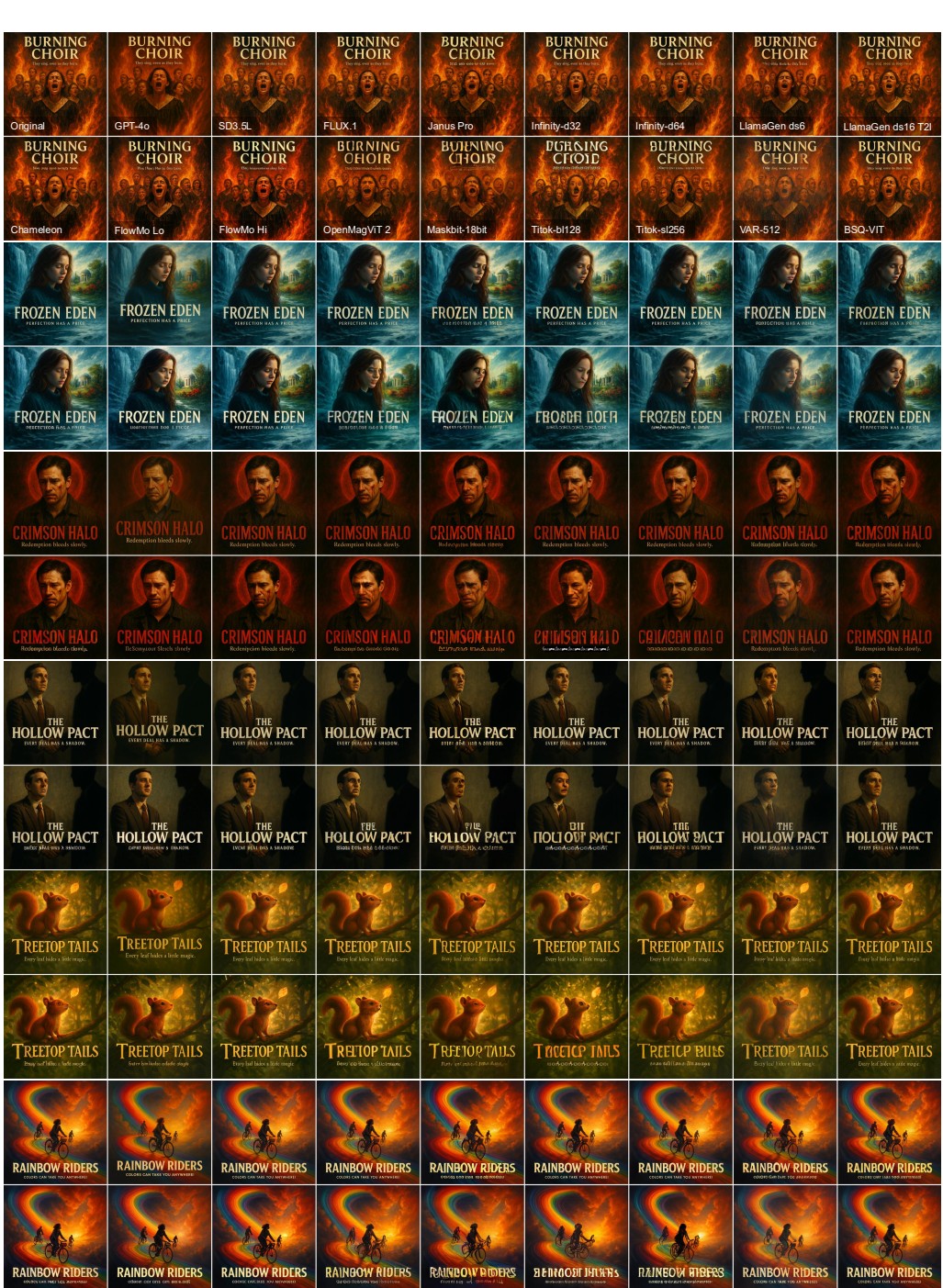

Figure 12: Additional visualize qualitative results for movie posters of task 3. The corresponding model names are listed in the first row.

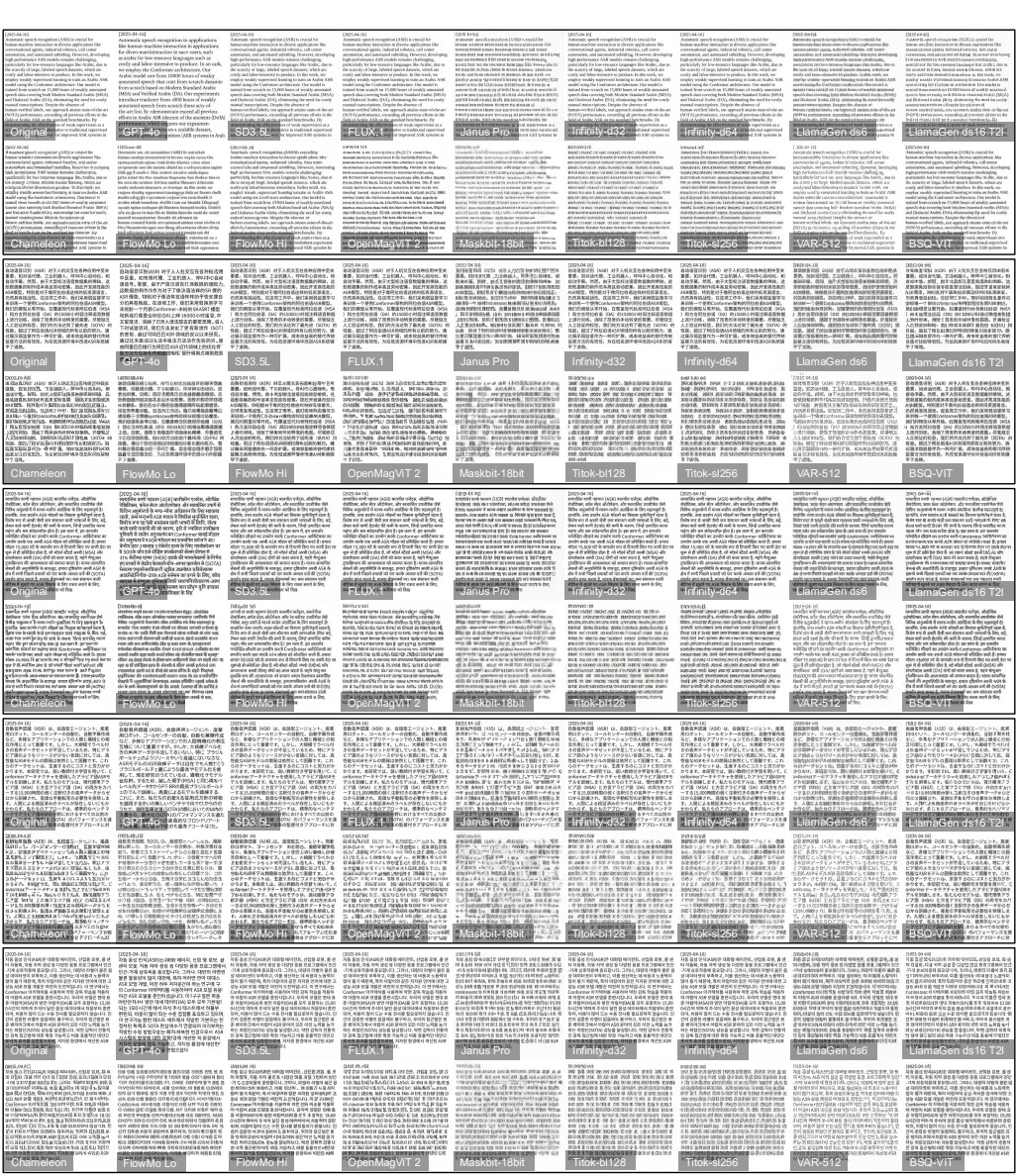

Figure 13: Additional qualitative results showing one ArXiv abstract and its corresponding multi-lingual versions (Chinese, Japanese, Korean, and Hindi) reconstructed by various visual tokenizers.

