# OpenReview forum: "VTBench: Evaluating Visual Tokenizer for Autoregressive Image Generation"
_ICLR.cc/2026/Conference — ICLR 2026 Conference Withdrawn Submission_

### Official Review · Reviewer_WWhR · 2025-10-19

**Soundness:** 4
**Presentation:** 4
**Contribution:** 4
**Rating:** 6
**Confidence:** 4

**Summary:**

The paper posits that the visual tokenizer (VT) is a primary performance bottleneck for autoregressive image generation models. The authors construct a new benchmark, VTBench, to systematically evaluate VTs in isolation, a task not covered by existing end-to-end generation benchmarks. VTBench comprises three tasks—Image Reconstruction, Detail Preservation, and Text Preservation—designed to stress-test VTs under diverse conditions (e.g., high/varying resolution, multilingual text). The experimental results, covering a wide range of VTs from recent models, demonstrate a consistent and significant performance gap between discrete tokenizers and continuous VAEs or the proprietary GPT-4o system. The work concludes by releasing the benchmark to spur community effort in developing improved open-source VTs.

**Strengths:**

1. Problem Formulation: The paper identifies and formalizes a critical, yet overlooked, evaluation gap in the literature. Isolating the VT component for analysis is a necessary step for principled progress in AR generative models, and this paper provides the first comprehensive framework to do so.
2. Benchmark Rigor: The proposed VTBench is thorough. Its multi-task structure, especially the inclusion of challenging sub-tasks like high-density academic text and non-Latin scripts, goes beyond standard ImageNet-based reconstruction and provides a much more demanding and realistic testbed for modern multimodal requirements.
3. Empirical Contribution: The extensive empirical study provides a clear and valuable snapshot of the current state of visual tokenization. The finding that virtually all open-source discrete VTs fail significantly on complex tasks is a strong, sobering result for the community.
4. Resource Provision: The public release of the benchmark, dataset, and codebase is a significant contribution that will lower the barrier for future research and enable standardized comparisons.

**Weaknesses:**

1. Analysis is Diagnostic, not Prescriptive: The paper provides an excellent diagnosis of the problem but offers limited insight into why certain designs fail and what specific architectural principles lead to success. The analysis primarily ranks models but does not deeply correlate architectural choices (e.g., quantization method, downsampling factor, use of hierarchy) with specific outcomes in VTBench. This leaves the reader understanding the problem but with little concrete guidance on how to build a better VT.
2. Potential for Dataset Bias: The use of GPT-4o to synthesize several of the more challenging evaluation sets (e.g., for detail and text preservation) introduces a potential methodological confounder. As the internal architecture of GPT-4o is unknown but hypothesized to be AR-based, the data it generates may have an inherent distributional bias that favors certain types of architectures over others. A discussion of this limitation is warranted. The evaluation tasks, particularly for text and detail, are performed in a "clean room" environment. The text is computer-rendered, and the images are either high-quality photographs or pristine synthetic generations. This overlooks the primary challenge of real-world vision: handling noise, degradation, and unpredictable variations (e.g., motion blur, compression artifacts, handwritten text, low lighting). A VT optimized for perfect reconstruction on clean data might be brittle and fail catastrophically on noisy, real-world inputs.
3. Lack of Human-in-the-Loop Evaluation: The paper relies exclusively on automated metrics. For tasks like Detail Preservation, metrics such as LPIPS are known to be imperfect proxies for human perceptual judgment. While understandable for a large-scale benchmark, a small-scale user study to validate whether the metric-based rankings align with human perception would have made the conclusions on reconstruction quality more robust.
4. Attribution of GPT-4o's Performance: The paper attributes GPT-4o's strong performance to a superior VT. While plausible, this is an inference. The performance could also stem from a powerful diffusion-based decoder that is exceptionally good at correcting tokenization artifacts, or a combination of both. The paper could be more precise by framing this as a hypothesis and discussing alternative explanations.

**Questions:**

1. From your extensive analysis, can you distill any specific design principles? For instance, does residual quantization (as in VAR/Infinity) consistently outperform other methods across all tasks, or does it excel in detail preservation at the cost of, say, text fidelity? A more fine-grained "if you need X, prioritize Y" analysis would be highly valuable.
2. Regarding the GPT-4o generated data: Could you discuss the potential for dataset bias? Have you considered validating the key findings from these sub-tasks on a small, curated set of real-world images (e.g., from technical manuals or artistic photography) to ensure the conclusions generalize beyond GPT-4o's output distribution?
3. You hypothesize that GPT-4o may use an RVAE-like tokenizer. If an open-source model were to adopt such an architecture, what, in your view, would be the most significant remaining challenge to closing the performance gap? Is it purely a matter of scale, or are there other fundamental architectural or training subtleties that VTBench might not capture?

---

### Official Review · Reviewer_tgCE · 2025-10-28

**Soundness:** 3
**Presentation:** 3
**Contribution:** 2
**Rating:** 2
**Confidence:** 4

**Summary:**

This paper introduces VTBench, a benchmark designed to evaluate the reconstruction accuracy of visual tokenizers used in autoregressive models. VTBench consists of three tasks: Image Reconstruction, Detail Preservation, and Text Preservation. Two evaluation metrics CER (Character Error Rate) and WER (Word Error Rate) are introduced to measure text reconstruction ability of visual tokenizers. Experimental results show that the reconstruction performance of publicly available visual tokenizers for autoregressive models remains inferior to that of continuous visual tokenizers used in diffusion models.

**Strengths:**

- The paper is well-written and easy to follow.
- The proposed metrics CER and WER can effectively measure the text reconstruction performance of visual tokenizers.
- Experiments covered a wide range of visual tokenizers used in autoregressive image generation.

**Weaknesses:**

- While detail preservation is an important aspect of visual tokenizers as highlighted in the paper, the evaluation metrics used for the detail preservation task in Table 2 remain traditional ones such as PSNR, SSIM, LPIPS, and FID. These metrics may not effectively capture the nuanced differences between visual tokenizers in their ability to preserve fine details.
- The dataset construction heavily relies on images generated by GPT-4o, including high-resolution images for Task 1, images for Task 2, and movie posters for Task 3. Although GPT-4o is a powerful image generation model, its outputs may not match the quality of real-world images, particularly in detailed structures and textual content. This reliance on generated data could introduce biases into the benchmark and potentially affect the validity of subsequent evaluations and developments.
- The technical contribution of the paper appears to be limited. The introduced Reconstruction task (Task 1) overlaps with existing benchmarks, and no new evaluation metrics are proposed for the newly introduced Detail Preservation task.
- While reconstruction accuracy is an important ability of visual tokenizers which sets the upperbound of generation models, differences in latent space structure can also significantly influence downstream generation quality and thus deserve further attention.

**Questions:**

Please refer to the weakness session.

---

### Official Review · Reviewer_Jsih · 2025-11-02

**Soundness:** 3
**Presentation:** 3
**Contribution:** 2
**Rating:** 4
**Confidence:** 3

**Summary:**

This paper presents VTBench, a benchmark for evaluating Visual Tokenizers (VTs) in Autoregressive (AR) image generation. It isolates VT performance through three tasks Image Reconstruction, Detail Preservation, and Text Preservation covering scenarios such as high/variable resolutions, text-rich layouts, and multilingual scripts. Experiments on various SOTA VTs show that discrete VTs lag far behind continuous VAEs in fidelity, detail, and text accuracy, especially under complex conditions.

**Strengths:**

The paper addresses a clear gap in AR image generation by introducing VTBench, the first dedicated benchmark for evaluating visual tokenizers. The benchmark design is systematic and covers diverse, realistic scenarios, including high-resolution, variable-resolution, and multilingual text inputs. The experimental evaluation is thorough, uses appropriate quantitative metrics (e.g., CER, WER), and provides insights into the limitations of current discrete VTs.

**Weaknesses:**

While VTBench is valuable, the paper scope is mostly limited to reconstruction metrics and high-level qualitative analyses; less emphasis is placed on why certain VT architectures fail and how they could be improved beyond generic suggestions. The benchmark depends heavily on GPT-4o-generated datasets, which may introduce biases and limit applicability in other domains.

**Questions:**

Since VTBench relies heavily on GPT-4o generated data, how do you assess potential bias in high-resolution or multilingual scenarios, and would results differ with real-world images? Could the dataset and evaluation be expanded with a more fine-grained categorization, for example, sub-categories within each task (e.g., different types of textures, object classes, or text styles)?

---

### Official Review · Reviewer_T18B · 2025-11-02

**Soundness:** 3
**Presentation:** 2
**Contribution:** 2
**Rating:** 4
**Confidence:** 3

**Summary:**

**Problem.** Autoregressive (AR) image generators hinge on the **visual tokenizer (VT)**, yet current VTs may lose structure, fine detail, and text, and most benchmarks don’t isolate VT quality. **Method.** The paper proposes **VTBench**, isolating VT performance via three tasks—Image Reconstruction, Detail Preservation, and Text Preservation—spanning ImageNet, 1024×1024 “high-res,” varying resolutions, and multilingual text; metrics include PSNR/SSIM/LPIPS/FID and OCR-based CER/WER (Fig. 3). **Key innovations.** A VT-only evaluation protocol with diverse conditions, a curated high-frequency detail set, and text-heavy (English/multilingual) suites. **Main results.** Across Tables 1–3 and qualitative figs., discrete VTs lag far behind continuous VAEs (e.g., FLUX.1-dev/SD3.5L) in reconstruction/detail/text; many discrete VTs fail on arbitrary resolutions. **Significance.** If adopted broadly, VTBench could clarify VT bottlenecks for AR pipelines and catalyze stronger, reusable open VTs.

**Strengths:**

- **Clear scope separation.** Evaluates the VT in isolation with well-defined tasks/metrics (Fig. 3; §3).
- **Comprehensive scenarios.** Includes 1024×1024 and varying resolutions; multilingual and dense-text images (Tables 1–3; Figs. 4–5).
- **Consistent, interpretable findings.** Continuous VAEs strongly outperform discrete VTs in fidelity and text preservation (Tables 1–3).

**Weaknesses:**

1) **Fairness across VT capacity/bit-rate is unclear.**
Table 1 mixes tokenizers with very different parameter counts and latent granularities (e.g., 54 M→946 M; multiple quantizers/codebooks) but does not **bit-rate-match** or control for tokens-per-image. Without rate control (bits/pixel) or matched FLOPs/latents, conclusions like “continuous VAEs are better” risk capacity confounds. Please add **bit-rate-controlled** comparisons and per-model tokens-per-image/FLOPs/latency. (Table 1; §4.1).

2) **High-resolution benchmark uses GPT-4o-generated images as “ground truth.”**
You synthesize 100× 1024² images with GPT-4o for Tasks 1–2 detail sets (and posters/abstracts in Task 3), but GPT-4o’s pipeline is undisclosed and may introduce **style/anti-aliasing biases** that favor continuous decoders. Please justify this choice with an **independent high-res source** (e.g., DIV2K/DF2K) and report sensitivity of results to the source generator. (Task definitions; §3.1–3.3).

3) **Resolution robustness conflates *acceptance windows* with quality.**
Table 1 marks many discrete VTs as “unsupported” at 1024 or variable size; Fig. 4 reports semantic failures when forced. This blends **architectural constraints** (fixed input grid) with **reconstruction quality**. Please separate: (a) acceptance (can the VT ingest the size?) vs. (b) quality conditional on legal sizes. Provide per-VT legal-size baselines and an **oracle resize** control. (Fig. 4; §4.1).

4) **Text preservation pipeline depends on a single OCR backend.**
CER/WER rely on a single OCR; while an appendix claims robustness, the main text lacks head-to-head with alternate OCRs or **human adjudication**. Please add PaddleOCR/Tesseract/TrOCR ablations and inter-annotator agreement for a subset; report κ/CI. (Table 3; §4.3).

5) **Metric choice may penalize slight misalignments.**
FID on reconstructions (not generations) is sensitive to resizing/codec choices; PSNR/SSIM are brittle to tiny shifts. You include LPIPS, which helps, but **DISTS/LPIPS-Alex/VGG variants** and **precision/recall** for texture content would strengthen claims. Please add these, plus **per-class** breakdowns for patterned/detail images. (Task 2; Table 2).

6) **Lack of variance and statistical testing.**
Core tables report single numbers; improvements are large for VAEs but smaller among discrete VTs (e.g., MaskBiT 16→18 bit, Titok variants). Add **μ±σ across ≥3 seeds**, paired tests, and **effect sizes**; include CIs for CER/WER. (Tables 1–3).

7) **Causality between VTBench and end-to-end AR quality is asserted, not shown.**
Introduction claims VTs “define the upper bound” of AR performance, but there’s no **correlation study** linking VTBench scores to downstream AR generation metrics across a fixed AR model. Please provide correlation (Spearman/Pearson) between VTBench and end-to-end GenEval/T2I-compbench on the same AR backbones. (Abstract/Intro; Fig. 1).

8) **Heterogeneous preprocessing & resizing policies.**
Task 1 mixes center-crop to model-specific size, fixed 1024², and “native” varying resolutions; different VTs may use different antialias filters. Please tabulate **exact resize/antialias** per VT and add a **shared canonical preproc** control. (Task 1; §3.1; Table 1).

9) **Use-case coverage is narrow for documents/UI.**
Text-heavy sets use posters/abstracts; real-world documents (scans, forms, receipts, UI screenshots) differ in fonts/noise. Please add at least one **document/UI** corpus and report CER/WER by font size and stroke width. (Task 3; Table 3).

10) **Reproducibility details need hard artifacts.**
You promise an anonymized repo and eventual open-sourcing; for review, please release **frozen configs, seeds, exact prompts, generated image hashes**, and scripts to re-mint the synthetic sets. (Ethics/Reproducibility).

11) **Interpretation of GPT-4o results is speculative.**
The paper hypothesizes AR aspects of GPT-4o; while interesting, this belongs in Appendix with clearer caveats and should not anchor the benchmark narrative. Please move speculation and add **ablation removing GPT-4o-sourced content** from Tasks 2–3 to test sensitivity. (Intro/Discussion).

**Questions:**

See Weaknesses

---

### Note · Authors · 2025-11-12

I have read and agree with the venue's withdrawal policy on behalf of myself and my co-authors.